



# Multi-scale snowdrift-permitting modelling of mountain snowpack

Vincent Vionnet[1,2], Christopher B. Marsh[1], Brian Menounos[3], Simon Gascoin[4], Nicholas E. Wayand[1], Joseph Shea[3], Kriti Mukherjee[3] and John W. Pomeroy[1].

[1]Centre for Hydrology, University of Saskatchewan, Saskatoon, Canada
[2]Environmental Numerical Prediction Research, Environment and Climate Change Canada, Dorval, QC, Canada
[3]Natural Resources and Environmental Studies Institute and Geography Program, University of Northern British Columbia, Prince George, V2N 4Z9, Canada
[4]Centre d'Etudes Spatiales de la Biosphère, UPS/CNRS/IRD/INRAE/CNES, Toulouse, France

*Correspondence to*: Vincent Vionnet (vincent.vionnet@canada.ca)

**Abstract.** The interaction of mountain terrain with meteorological processes causes substantial temporal and spatial variability in snow accumulation and ablation. Processes impacted by complex terrain include large-scale orographic enhancement of snowfall, small-scale processes such as gravitational and wind-induced transport of snow, and variability in the radiative

balance such as through terrain shadowing. In this study, a multi-scale modeling approach is proposed to simulate the temporal and spatial evolution of high mountain snowpacks using the Canadian Hydrological Model (CHM), a multi-scale, spatially distributed modelling framework. CHM permits a variable spatial resolution by using the efficient terrain representation by unstructured triangular meshes. The model simulates processes such as radiation shadowing and irradiance to slopes, blowing snow redistribution and sublimation, avalanching, forest canopy interception and sublimation and snowpack melt. Short-term,

km-scale atmospheric forecasts from Environment and Climate Change Canada's Global Environmental Multiscale Model through its High Resolution Deterministic Prediction System (HRDPS) drive CHM, and were downscaled to the unstructured mesh scale using process-based procedures. In particular, a new wind downscaling strategy combines meso-scale HRDPS outputs and micro-scale pre-computed wind fields to allow for blowing snow calculations. HRDPS-CHM was applied to simulate snow conditions down to 50-m resolution during winter 2017/2018 in a domain around the Kananaskis Valley (~1000

km[2]) in the Canadian Rockies. Simulations were evaluated using high-resolution airborne Light Detection and Ranging (LiDAR) snow depth data and snow persistence indexes derived from remotely sensed imagery. Results included model falsifications and showed that both blowing snow and gravitational snow redistribution need to be simulated to capture the snowpack variability and the evolution of snow depth and persistence with elevation across the region. Accumulation of wind-blown snow on leeward slopes and associated snow-cover persistence were underestimated in a CHM simulation driven by

wind fields that did not capture leeside flow recirculation and associated wind speed decreases. A terrain-based metric helped to identify these lee-side areas and improved the wind field and the associated snow redistribution. An overestimation of snow redistribution from windward to leeward slopes and subsequent avalanching was still found. The results of this study highlight





the need for further improvements of snowdrift-permitting models for large-scale applications, in particular the representation of subgrid topographic effects on snow transport.

# 1 Introduction

High mountain snowpacks are characterized by a strong spatial and temporal variability that is associated with elevation, vegetation cover, slope steepness, orientation and wind exposure. This variability results from processes occurring during the snow accumulation and ablation periods at a large range of spatial scales (e.g., Pomeroy and Gray, 1995; Pomeroy et al., 1998; 2012; 2016; Clark et al., 2011; Mott et al., 2018). Snow accumulation at the mountain range scale (1-500 km) is primarily dominated by orographic precipitation, and results in regions of enhanced or reduced snowfall (e.g., Houze, 2012). At the mountain-ridge and slope scales (5 m – 1 km), preferential deposition of snowfall and wind-induced snow transport strongly impact snow accumulation (e.g., Mott et al. 2018). Redistribution by avalanches (e.g., Bernhardt and Schulz, 2010; Sommer et al., 2015) and surface and blowing snow sublimation (e.g., MacDonald et al., 2010; Vionnet et al., 2014; Musselman et al., 2015; Sextone et al., 2018) also modify the spatial variability of snow. During the ablation period, spatially varying melt rates result from differences in solar irradiance due to aspect and shading (e.g., Marks and Dozier, 1992; Marsh et al., 2012), in net solar irradiance due to albedo variations (e.g., Dumont et al., 2011; Schirmer and Pomeroy, 2020), in turbulent fluxes (e.g., Winstral and Marks, 2014; Gravelman et al., 2015) and in advected heat from snow-free ground in patchy snow cover conditions (e.g., Mott et al., 2013; Harder et al., 2017).

The multi-scale variability of mountain snow represents a challenge for snow models used in support of avalanche hazard forecasting (Morin et al., 2020), hydrological predictions (e.g., Warscher et al., 2013; Brauchli et al., 2017; Freudiger et al., 2017) and climate projections (e.g., Rasouli et al., 2014; Hanzer et al., 2018) in mountainous terrain. Several modelling strategies have been proposed to face this challenge and to capture this multi-scale variability. At the mountain range scale, atmospheric models at sufficient resolutions (4-km or finer) can bring valuable information on the variability of snowfall and resulting snow accumulation (e.g., Prein et al., 2015; Lundquist et al., 2019; Fang and Pomeroy, 2020). Indeed, at these resolutions, atmospheric models operate at convection-permitting scales, and explicitly represent convection and highly resolved vertical motions, achieving improved estimates of snowfall (e.g., Rasmussen et al., 2011). Sub-grid parameterization of snow depth have been proposed to represent the snow variability at the mountain-ridge and slope-scale for snowpack models operating at km scales (Liston, 2004; Helbig and van Herwijnen, 2017; He and Ohara, 2019). Another strategy consists of explicitly modelling the snow evolution at the mountain-ridges and slopes scales at resolutions ranging from a few meters to 200 meters (Liston, 2004; Musselman et al., 2015). At theses scales, the variability of snow accumulation can be represented using (i) simple parameterizations to adjust snowfall as a function of topographic parameters (e.g., Winstral and Marks, 2002; Hanzer et al., 2016) or (ii) using models that explicitly represent preferential deposition and/or wind-induced snow redistribution (e.g., Essery et al., 1999; Durand et al., 2005; Pomeroy et al., 2007; Liston et al., 2007; Lehning et al., 2008; Sauter et al., 2013; Vionnet et al., 2014; Marsh et al., 2020a). These models can be classified as snowdrift-permitting models





since they operate at sufficient resolutions (200-m or finer) to activate the horizontal redistribution of snow between computational elements. High resolution remote sensing data assimilation can also be used at these scales to correct spatial biases in the atmospheric forcing and to account for missing physical processes in the models (e.g., Durand et al., 2008; Baba et al., 2018).

Snowdrift-permitting models simulate wind-induced snow transport in the saltation and suspension layers (e.g, Pomeroy and
Gray, 1995). As proposed by Mott et al. (2018), they can be divided into two main categories: (i) models solving the vertically-integrated mass flux in the saltation and suspension layers (Essery et al., 1999; Durand et al., 2005; Pomeroy et al., 2007; Liston et al., 2007) and (ii) models solving the three-dimensional (3-D) advection-turbulent diffusion equation of blown snow particles in the atmosphere (Gauer, 1998; Lehning et al., 2008; Schneiderbauer and Prokop, 2011; Sauter et al., 2013; Vionnet et al., 2014). One of the main challenges for all these models is obtaining accurate driving wind fields at sufficient high
resolution since they strongly impact the accuracy of simulated snow redistribution (Mott and Lehning, 2010; Musselman et al., 2015).  Models of the first category need two-dimensional (2-D) driving wind fields. Liston et al. (2007) inspired by Ryan (1977) proposed the use of terrain-based parameters to adjust distributed wind fields to the local topography. These distributed wind fields can be obtained from interpolated station data (Gascoin et al., 2013; Sextone et al., 2018), hourly output from regional climate model at convective-permitting scale (Reveillet et al., 2020), or a pre-computed wind field library using an
atmospheric model (Berhnardt et al., 2010). Essery et al. (1999) used a linearized turbulence model (Walmsley et al., 1982) to build a pre-computed library of 2-D wind maps to distribute wind measurements from stations data. Musselman et al. (2015) showed that this approach led to more accurate simulations of snow redistribution around an alpine crest than wind fields derived from the terrain-based parameters proposed by Liston et al. (2007). Models of the second category require a 3-D representation of the wind field and associated atmospheric turbulence. In this case, driving wind fields can be obtained from
computational fluid dynamics (CFD) models (Gauer, 1998; Schneiderbauer and Prokop, 2011), or atmospheric models in Large Eddy Simulations (LES) mode used to generate library of pre-computed wind fields (Lehning et al., 2008; Mott and Lehning, 2010) or fully coupled to a snowpack model (Vionnet et al., 2014). These advanced models can be used for detailed studies such as the feedbacks between blowing snow sublimation and the atmosphere (Groot Zwaaftink et al., 2011) or the processes driving the variability of snow accumulation during a snowfall event (Vionnet et al., 2017).

Differences in the level of complexity of snowdrift-permitting models and associated driving wind fields influence the spatial and temporal ranges of application of these models. Due to their relatively low computational costs, models of the first category can be applied to simulate the snow cover evolution over entire snow seasons at resolution between 25 and 100 m for regions covering hundreds of km$^2$ (e.g., 210 km$^2$ for Berhnardt et al. (2010); 1043 km$^2$ for Gascoin et al. (2013); 3600 km$^2$ for Sextone et al. (2018)). On the other hand, models of the second category are usually restricted to the simulation of single blowing snow
events at resolution between 2 m and 50 m over regions covering tens of km$^2$ (e.g., 1 km$^2$ in Schneiderbauer and Prokop (2011), 2.3 km$^2$ in Mott and Lehning (2010); 23 km$^2$ in Vionnet et al. (2017)). The study by Groot Zwaaftink et al. (2013) is an exception and relied on the Alpine 3D model (Lehning et al., 2008) to simulate the snow cover evolution at 10-m resolution over a region of 2.4 km$^2$ of the Swiss Alps for an entire winter. All these snowdrift-permitting models used a gridded



representation of the topography. Large-scale applications of these models over mountainous area are limited by the need to
have a fixed and sufficiently high resolution over large areas even in regions where wind-induced snow transport is not active
(valley bottom for example).

To overcome some of these limitations, Marsh et al. (2020a) developed a snowdrift-permitting scheme of intermediate
complexity that solves the 3-D advection-diffusion blowing snow transport on a variable resolution unstructured mesh. This
scheme is implemented in the Canadian Hydrological Model (CHM; Marsh et al., 2020b). The landscape is discretized using
a variable resolution unstructured mesh that allows an accurate representation of terrain heterogeneities with limited
computation elements (Marsh et al., 2018). Marsh et al. (2020a) used the WindNinja diagnostic wind model (Forthofer et al.,
2014) to build libraries of pre-computed wind fields. Wagenbrenner et al. (2016) showed that WindNinja can be used to
downscale wind field from atmospheric models running at convection-permitting scale in complex terrain.

The objective of this study is to present and evaluate a novel strategy for multi-scale modelling of mountain snowpack over
large regions and for entire snow seasons. Specifically: (1) Can efficient wind-downscaling approaches be used for blowing
simulation? (2) Over large spatial extents, can lateral mass redistribution (blowing snow and avalanching) be ignored? (3) Can
optical satellite imagery be used to diagnose model performances over large spatial extents? This modelling strategy combines
(i) atmospheric forcing from the convection-permitting Canadian Numerical Weather Prediction (NWP) system, (ii) a
downscaling module including wind fields from a high-resolution diagnostic wind model and (iii) the multi-scale snowdrift-
permitting model CHM running on an unstructured mesh. This modelling strategy was applied for a full winter over a domain
of 958 km$^2$ around the Kananaskis Valley in the Canadian Rockies. Different model configurations were tested to assess the
impact of the representation of physical processes in CHM as well as the complexity of the wind downscaling scheme. Airborne
LiDAR snow depth data and snow persistence indexes derived from Sentinel-2 images were used to evaluate the ability of the
different CHM configurations to capture the elevation-snow depth relationship as well as snow redistribution around wind-
exposed ridges. The paper is organized as follows: Section 2 presents the study area and the different observation datasets used
in this study, and also describes the CHM modelling platform, the wind downscaling strategy and the configurations of the
CHM experiments; Section 3 evaluates the impact of the wind field downscaling and the quality of the snowpack simulations
using airborne LiDAR snow depth data and snow persistence indexes; Section 4 discusses the main challenges associated with
snowdrift-permitting modelling of mountain snowpack and associated limitations. Finally, concluding remarks are presented
in Section 5.

## 2 Data and Methods

### 2.1 Study Site

This work studies the evolution of the mountain snowpack around the Kananaskis Valley of the Canadian Rockies, Alberta
(Fig. 1). The study domain covers an area of 958 km$^2$ and is characterized by a complex and rugged topography with elevations
ranging from 1400 m a.s.l. at the Kananaskis valley bottom in the northeastern part of the domain up to 3406 m a.s.l. at the





summit of Mount Sir Douglas in the southern part of the region (Fig. 1b). Valley bottoms and lower slopes are predominately covered by needleleaf evergreen forest (Fig. 1a). Short shrubs and low vegetation are present near treeline whereas exposed rock surfaces, talus and grasses are found in the highest alpine elevations. The Kananaskis Valley hosts several meteorological stations that are part of the University of Saskatchewan's Canadian Rockies Hydrological Observatory (CRHO;

https://research-groups.usask.ca/hydrology/science/research-facilities/crho.php) and is active for research in snow hydrology (e.g., MacDonald et al., 2010; Musselman et al., 2015; Pomeroy et al., 2012; 2016; Fang et al., 2019; Fang and Pomeroy, 2020). More details about these meteorological stations are given in Sect. 2.3.1.

## 2.2 Model

### 2.2.1 Mesh generation

The Digital Elevation Model (DEM) from the Shuttle Radar Topography Mission-SRTM (EROS Center, 2017) at a resolution of 1-arc second (30 m) was used as input to the *mesher* code (Marsh et al., 2018) to generate an unstructured, variable resolution triangular mesh over the Kananaskis domain (Fig. 1). In *mesher*, triangles are bounded with minimum and maximum areas and are generated to fulfil a given tolerance defined here as the root mean square error to the underlying topographic raster. This study uses a high-resolution mesh, denoted $M^{15}_{50}$, with a minimum triangle area of 50 m × 50 m and a vertical tolerance

of 15 m. The characteristics of the generated mesh are given in Table 1. For the Kananaskis domain, 383 200 raster grid cells with a 50 m resolution are required to represent the terrain, whereas 101 700 triangles are used in $M^{15}_{50}$ (Fig. 2). Large triangles are found in valley bottoms of low topographic variability, whereas small triangles dominate in alpine terrain, close to ridges where wind-induced snow redistribution is common.

A dataset of tall vegetation (>5 m) coverage, with a resolution of 30 m (Fig. 1a), was obtained from Hansen et al. (2013).

These fractional values were applied to the triangular mesh via *mesher* by averaging the raster cells that correspond to each triangle and assigning this average to the triangle. Triangles with an average fraction of high-vegetation larger than 0.5 were classified as forest.

### 2.2.2 Snowpack model

Distributed snowpack simulations over the triangular mesh of the study area were performed using the version of the Snobal

scheme (Marks et al. 1999) implemented in CHM (Marsh et al. 2020b). Snobal has been used in numerous mountainous regions across North America (e.g., Garen and Marks, 2005; Pomeroy et al., 2016; Hedrick et al., 2018). Snobal is a physically based snowpack model that approximates the snowpack with two layers. The surface layer was implemented here with a fixed thickness of 0.1 m and is used to estimate surface temperature for outgoing longwave radiation and turbulent heat fluxes. The second lower layer represents the remaining snowpack. For each layer, Snobal simulates the evolution of the snow water

equivalent (SWE), temperature, density, cold content, and liquid water content. The version of Snobal used in this study includes an improved algorithm for snow compaction that accounts for bulk compaction and temperature metamorphism



(Hedrick et al., 2018). Snobal in CHM employs the snow albedo routine of Verseghy et al. (1993). The ground heat flux assumes heat flow to a single soil layer of known temperature and thermal conductivity. In these simulations, the soil temperature was set to -4°C at 10 cm below the soil-snow interface. Marsh et al. (2020b) used the same value for Snobal simulations with CHM at the Marmot Creek Research Basin located further north in the Kananaskis Valley (Fig. 1).

CHM also includes a 3-D advection-diffusion blowing snow transport and sublimation model (Marsh et al., 2020a). This scheme uses a finite volume method discretization on the unstructured mesh. It deploys the parameterization of Li and Pomeroy (1997) to determine the threshold wind speed for snow transport initiation as a function of air temperature. In case of blowing snow occurrence, the steady-state saltation parameterization of Pomeroy et al. (1990) is used to compute the mass concentration in the saltation layer. The concentration in the saltation layer is impacted by shear stress partitioning due to the presence of vegetation (such as shrubs) and the upwind fetch. Upwind fetch is calculated for each triangle of the mesh using the *fetchr* parameterization of Lapen and Martz (1993). The saltation layer acts as a lower boundary condition for the suspension layer, which is discretized with a user-defined number of layers to resolve the gradient of concentration of blowing snow particles in the suspension layer. For each layer, PBSM-3D solves the evolution of the concentration of blowing snow particles accounting for advection, turbulent diffusion, sedimentation and mass loss due to sublimation based on the parameterizations proposed by Pomeroy and Male (1992) and Pomeroy et al. (1993). At a given time step, erosion and deposition rates are computed as the spatial divergence of the saltation and suspension fluxes and the snowpack simulated by Snobal is updated accordingly. In this study, 10 layers were used for a total height of the suspension layer of 5 m as in Marsh et al. (2020a).

In steep alpine terrain, gravitational snow transport strongly affects the spatial variability of the snowpack (e.g. Sommer et al., 2015) and modifies the runoff behaviour of alpine catchments (Warscher et al. 2013). For these reasons, the SnowSlide scheme (Bernhard and Schulz, 2010) was implemented in CHM. SnowSlide is a simple topographically driven model that simulates the effects of gravitational snow transport. SnowSlide uses a snow holding depth that decreases exponentially with increasing slope angle, limiting snow accumulation in steep terrain. SnowSlide was initially developed for regular gridded rasters and has been adapted here to the unstructured triangular mesh used by CHM. SnowSlide operates from the highest triangle of the mesh to the lowest one. If the snow depth exceeds the snow holding capacity for a given triangle, excess snow is redistributed to the lower adjacent triangles, proportionally to the elevation difference between the neighboring triangles and the original one. SnowSlide uses the total elevation (snow depth plus surface elevation) to operate. In this study, the default formulation of the snow holding depth proposed by Bernhardt and Schulz (2010) is used which leads to a maximal snow thickness (taken perpendicular to the slope) of 3.08 m, 1.11 m, 0.45 m, and 0.15 m for slopes of 30°, 45°, 60°, and 75°, respectively.

The impact of the presence of forest vegetation on snow interception, sublimation, snowpack accumulation and melt energetics are represented in CHM using the same canopy module as in the Cold Region Hydrological Model (CRHM; Ellis et al., 2010; Pomeroy et al. 2012). This module used Leaf Area Index and canopy closure to compute the effect of forests on shortwave and longwave irradiance at the snow surface. Snow interception and sublimation of intercepted snow are also represented following Hedstrom and Pomeroy (1998). In this study, the canopy module was activated for the triangles covered by forest as described in Sect. 2.2.1.



### 2.2.3 Atmospheric forcing

Snobal and PBSM-3D require the following atmospheric forcing: air temperature, humidity, wind speed, wind direction, liquid and solid precipitation rates, and longwave and shortwave irradiance. Due to the scarcity of the network of meteorological stations in the region (Fig. 1), hourly atmospheric forcings were obtained from the High-Resolution Deterministic System

(HRDPS; Milbrandt et al., 2016). HRDPS is the high-resolution NWP system running the Global Environmental Multiscale Model (GEM) operationally over Canada at 2.5-km grid spacing. Successive HRDPS forecasts from the 00 and 12 UTC analysis time at 6 to 17 h lead time were extracted over the region and combined together to generate a continuous atmospheric forcing. Previous studies have also used distributed forcing data from NWP systems to drive snowpack models in mountainous terrain since these data can often represent the complex interactions between topography and atmospheric flow better than

sparse meteorological measurements (Quéno et al., 2016; Vionnet et al., 2016; Havens et al. 2019; Lundquist et al., 2019; Fang and Pomeroy, 2020).

The HRDPS atmospheric forcing at 2.5-km grid spacing was downscaled to the triangles of the CHM mesh. Horizontal interpolation was first applied using inverse-distance weighting from the closest four HRDPS grid points. Corrections for elevation differences were then applied to adapt the HRDPS meteorological forcing to the high-resolution topography of the

CHM mesh. Constant monthly lapse rates were used to adjust HRDPS 2-m air temperature and humidity (Kunkel, 1989; Shea et al., 2004). HRDPS temperature was reduced (increased) if the elevation of the triangle is higher (lower) than the elevation of the HRDPS grids points. Precipitation amounts were not modified to account for elevation difference as it was assumed that HRDPS already captures the main orographic effects affecting mountain precipitation (Lundquist et al., 2019). A cosine-correction was then applied to adjust precipitation falling on inclined triangle for mass-conservation purpose (Kienzle, 2011).

Downscaled temperature and humidity were finally used to compute the precipitation phase with the psychrometric energy balance method of Harder and Pomeroy (2013) that performed well in the Kananaskis Valley. Direct and diffuse solar irradiance were taken from the HRDPS forecast and direct irradiance was corrected for slope and aspect as described in Marsh et al. (2012). Local terrain shadowing and its impact on shortwave irradiance were calculated using the algorithm of Dozier and Frew (1990) adapted for unstructured meshes as described in Marsh et al. (2020b). Longwave irradiance was adjusted for

elevation difference using the climatological lapse rate of Marty et al. (2002). Finally, wind speed and direction were taken from the lowest HRDPS prognostic level at 40 m above the surface and were downscaled to the CHM mesh using the strategy described in the next section.

### 2.2.4 Wind field downscaling

Mountain wind fields are notoriously difficult to observe and model (Davies et al., 1995), and obtaining high-resolution wind

fields constitutes one of the greatest challenges for blowing snow models in mountainous terrain (e.g., Mott and Lehning, 2010; Vionnet et al., 2014; Musselman et al., 2015, Réveillet et al., 2020). In the context of this study, hourly HRDPS near-surface wind fields at the 2.5 km scale were downscaled to the CHM mesh over the full duration of the simulations (one water



year). This required a computationally efficient wind downscaling method. Therefore, the wind downscaling strategy used in this study was derived from the method proposed by Barcons et al. (2018) for mesoscale-to-microscale downscaling of near-
surface wind fields. This method combines precomputed microscale simulations with a mesoscale forecast using transfer functions. In their study, Barcons et al. (2018) combined the Weather Research and Forecast mesoscale model at 3-km grid spacing and the Alya-CFDWind microscale model at 40-m grid spacing. In our study, microscale wind simulations were generated with the WindNinja model. WindNinja is a mass-conserving diagnostic wind model, primarily designed to simulate mechanical effects of terrain on the flow (Forthofer et al., 2014). Forthofer et al., (2014) showed that the model captures
important terrain-induced flow features, such as ridgetop acceleration or terrain channeling and can improve wildfire spread predictions in complex terrain. Wagenbrenner et al. (2016) used the model to downscale near-surface wind forecast from NWP systems in complex terrain.

The application and extension of the Barcons et al (2018) approach for use on an unstructured mesh and to account for direction perturbations is detailed below. First, to build the windmap library, WindNinja was run at 50-m resolution over the Kananaskis
domain (Fig. 1). As WindNinja uses a regular grid, the input topography was taken from the same SRTM DEM at 30-m grid spacing that was used to build the CHM mesh (Sect. 2.2.1). WindNinja used a spatially constant, bare-earth roughness length ($z_0 = 0.01$ m) and vegetation effects were introduced later in the downscaling procedure, as described below. WindNinja simulations were carried out for 24 initial wind directions (each 15°) with an initial wind speed at 40 m above the surface set to 10 m s$^{-1}$. The height of 40 m corresponds to the lowest HRDPS prognostic level.
Then, for each wind direction in the windmap library, the transfer function $f$ was computed for use in the downscaling procedure given as:

$$f = \frac{U_{WN}}{< U_{WN} >_L} \qquad (1)$$

where $U_{WN}$ is the local wind speed ($U_{WN} = \sqrt{u_{WN}^2 + v_{WN}^2}$), $u_{WN}$ and $v_{WN}$ are the horizontal components of the wind at 50-m resolution, and $< U_{WN} >_L$ is the spatial average of $U_{WN}$ over an area of size $L×L$. By construction, when $L$ tends towards 0, $f$ tends towards 1. As $L$ increases, $f$ incorporates the local wind fluctuation induced by the micro-scale terrain features (Barcons
et al., 2018). A value of $L = 1000$ m was used in this study in agreement with the finding of Barcons et al. (2018) in complex terrain. Note that Barcons et al. (2018) used a circle instead of a square to compute the spatial average of the wind speed. Thus, $f$ acts as a speedup/slowdown factor that accounts for topographic impacts on wind speed.

To account for impacts on direction, the following approach was taken. The rasters of the windmap library containing the horizontal $u$ and $v$ wind components and the transfer function $f$ for each initial wind direction were applied to the triangles of
the unstructured mesh using the *mesher* code (Marsh et al., 2018). At each CHM time step, the HRDPS $u_{HRDPS}$ and $v_{HRDPS}$ wind components were spatially interpolated to the triangles centres with an inverse-distance interpolant using the four closest HRDPS grid points. For each triangle, the interpolated HRDPS wind direction, $\theta_{HRDPS}$, was then reconstructed from the interpolated HRDPS wind components, $u_{HRDPS\_int}$ and $v_{HRDPS\_int}$. This direction was used to select the two sets of precomputed



micro-scale wind components or the wind directions $\varphi_1$ and $\varphi_2$ that bounds $\theta_{HRDPS}$ (i.e., $\varphi_1 < \theta_{HRDPS} < \varphi_2$). These selected

microscale wind components including the local terrain effect were then linearly interpolated and recombined to obtain the downscaled wind direction $\theta_{Down}$. The transfer functions corresponding to the wind directions $\varphi_1$ and $\varphi_2$ were also linearly combined to obtain the final transfer function, $f_{down}$. It was finally applied to scale the modulus of the interpolated HRDPS wind speed and derive the final downscaled wind speed as in Barons et al. (2018):

$$U_{Down} = f_{Down}\sqrt{u^2_{HRDPS\_int} + v^2_{HRDPS\_int}} \qquad (2)$$

Wind speeds were then adjusted to 10-m wind speed using a logarithmic law and modified to include vegetation interactions

using the vegetation cover of the triangle as defined in Sect 2.2.1.

Forthofer et al. (2014) and Wagenbrenner et al. (2016, 2019) have shown that the mass-conserving version of WindNinja has difficulties simulating lee-side recirculation where flow separation occurs. This difficulty is due to the absence of a momentum equation in the WindNinja flow simulation (Forthofer et al., 2014). As lee-side flow strongly influences snow accumulation (e.g. Gerber et al., 2018), an additional and optional step was added to the wind downscaling procedure described above. It

consisted of a modification of the transfer functions $f_{down}$ to reduce wind speed in leeward areas prone to flow separation. At each CHM time step, leeward areas were identified using the Winstral topographic parameter $Sx$ (Winstral and Marks, 2002; Winstral et al., 2017), computed at each triangle using the downscaled wind direction, $\theta_{Down}$. The $Sx$ algorithm examines all triangles along a fixed search line emanating from the triangle of interest to determine which triangle has the greatest upward slope relative to the triangle of interest. Positive $Sx$ values indicate sheltering features whereas negative $Sx$ values indicate that

the triangle of interest height is the highest cell along the search line and is topographically exposed. In this study, the $Sx$ algorithm used a search distance of 300 m, as in Winstral et al. (2017). Triangles with $Sx$ values larger than $20°$ were considered susceptible to flow separation in agreement with previous studies on the onset of flow separation in complex terrain (e.g., Wood, 1995). For these triangles, the transfer function, $f_{down}$ was set to a value of 0.25 (Winstral et al., 2009). Note finally that a mass and momentum-conserving version of WindNinja is also available (Wagenbrenner et al., 2019). Wagenbrenner et al.

(2019) have shown that momentum-conservation improved flow simulation at windward and leeward locations compared to the mass-conserving version but numerical instabilities made this version of the code unusable in the complex topography of the Canadian Rockies.

### 2.2.5 Model experiments

A set of CHM experiments were designed to assess the effect of the wind field downscaling, and the impact of process

representation on snowpack simulations at snowdrift-permitting scales (Table 2). A reference CHM configuration including wind downscaling accounting for recirculation, and gravitational and blowing snow redistribution was first defined (*BS Av Rc*). A stepwise model falsification was then used, removing the following processes from the model: (i) recirculation effects in the wind downscaling procedure (*BS Av NoRc*), (ii) blowing snow redistribution with PBSM-3D (*NoBS Av*), (iii) gravitational snow redistribution with SnowSlide (*NoBS NoAv*), (iv) wind downscaling with WindNinja (*No Down*). Note that



all the CHM experiments considered in this study account for the effects of terrain slope and aspect on incoming shortwave radiation. These simulations covered the period from 1$^{st}$ September 2017 to August 31$^{st}$ 2018 to fully capture snow accumulation and ablation in the region. For each experiment, CHM outputs were rasterized to a 50 m x 50 m raster for model evaluation. This rasterization was done via the GDAL rasterization capabilities (GDAL/OGR contributors, 2020). In short, this algorithm takes the triangle geometry in conjunction with an output raster (with given cell sizes and domain extent) and

resolves which raster cells correspond to each triangle. On the case that two triangles share an output cell, an overwrite is used by the algorithm. The 50 m x 50 m area was selected as it corresponds to the minimal triangle area for high-resolution used in this study (Table 1).

## 2.3 Data and evaluation methods

### 2.3.1 Meteorological observations

Hourly meteorological data collected at CRHO stations were used to evaluate the precipitation and wind fields driving CHM (Table 3). These stations include those in Marmot Creek Research Basin (Fang et al., 2019) and Fortress Mountain Snow Laboratory (Harder et al., 2016) (Table 3 and Fig. 1); covering an elevation range from 1492 m to 2565 m. Table 3 also provides the Topographic Position Index (TPI) at the position of the stations (Table 3) as this metric provides a quantification of each station's elevation relative to its surrounding. In this study, TPI was defined as in Winstral et al. (2017) and consists

of the difference between each station's elevation on a 50-m raster minus the mean of all pixel elevations located within a 2-km radius from the station. Hourly meteorological data were obtained from quality-controlled 15-min observations using the same method as in Fang et al. (2019). In particular, solid precipitation data were corrected from wind-induced undercatch using the method proposed by Smith (2007). Simulated wind speeds were corrected to the sensor height of each station (including snow depth) using a standard log-law for the vertical profile of wind speed near the surface and an aerodynamic roughness of

1 mm typically found in snow-covered alpine terrain (e.g., Naaim Bouvet et al., 2010).

### 2.3.2 Airborne LiDAR snow depth data

Airborne laser scanning (ALS) surveys were performed over the Kananaskis region on 5 October, 2017 (late summer scan) and on 27 April 2018 (winter scan) using a Riegl Q-780 infrared (1024 micron) laser scanner with a dedicated Applanix POS AV Global Navigation Satellite System (GNSS) inertial measurement unit (IMU). The Q-780 scanner was flown at heights of

approximately 2500 m above the terrain that yielded swath widths of 2000 m to 3000 m. Post-processing of the ALS survey flight trajectory yielded vertical and horizontal positional uncertainties of ±15 cm (1σ). Post-processed point clouds data were exported into LAS files, and LAStools (https://rapidlasso.com/lastools/) was used to generate 5 m resolution digital elevation models (DEMs). The summer and winter DEMs were co-registered to minimize slope and aspect-induced errors (Nuth and Kääb, 2011). Additional details about the processing workflow over snow-covered terrain can be found in Pelto et al. (2019).

To estimate uncertainties on the snow depth retrieval, snow-free areas that included peaks and road surfaces were identified in





a 3-m satellite imagery (Planet Scope) for 27 April 2018. Analysis of elevation change over these snow-free surfaces (34 comparison points across all elevation) indicated an average (median) elevation change of -4.1 cm (0.5 cm) and a standard deviation of 19.8 cm. The median absolute deviation reached 8.0 cm. The DEM of snow depth was masked to only include non-glacierized terrain (Fig. 1b) and to exclude any areas of elevation change that was less than 0 m and greater than 20 m;

elevation change beyond these values are considered outliers (Grunewald et al., 2014) and can arise from steep terrain that was effectively in the shadow of the laser scanner. For model evaluation, the 5-m snow depth map was then resampled over the same 50-m raster as the CHM output, taking for each cell of the 50-m raster the average of all non-masked cells in the 5-m snow depth map. Cells of the 50-m raster that contained more than 75 % of masked cells in the 5-m snow depth map were masked out. In addition, grid points covered by glaciers identified in the Randolph Glacier Inventory (Pfeffer et al., 2014) were

removed from the analysis since elevation change over these surfaces is also influenced by ice dynamics (Pelto et al., 2019). Finally, forested pixels identified using the global database of Hansen et al. (2013) at 30-m grid spacing were masked out as well since this study focuses on snow redistribution processes in open terrain.

The distributions of simulated and observed snow depths were compared for different 200-m elevation bands for three sub-areas of the Kananaskis domain (Fig. 1b): (i) Kananaskis North; (ii) Kananaskis South and; (iii) Haig. These three sub-areas

were characterized by different mean (standard deviation) observed snow depths: 0.90 m (0.82 m) for Kananaskis North, 1.32 m (1.03 m) for Kananaskis South and 2.00 m (1.33 m) for Haig. For each elevation band, the Root Mean Squared Error (RMSE) and the Wasserstein distance of order 1, $W_1$, (Rüschendorf, 1985) were used to quantify the agreement between the simulated and the observed distributions. $W_1$ is defined as:

$$W_1(s,o) = \int_{-\infty}^{+\infty} |S(s) - O(o)|$$

where $s$ and $o$ are the simulated and observed snow depth distributions and $S$ and $O$ the corresponding cumulative distribution functions. $W_1$ has the same unit as the variable considered (here m for snow depth) and a perfect match between the distribution lead to $W_1 = 0$. For each sub-area, simulated and observed snow depth distributions were also compared as a function of slope orientation in the upper slopes using bias and $W_1$ to provide a specific assessment of model performances in regions particularly exposed to wind-induced snow-transport. Upper slopes in the 50-m raster were identified using the TPI as defined above.

Regions with TPI greater than 150 m were classified as upper slopes.

### 2.3.3 Sentinel-2 snow cover maps.

Wayand et al. (2018) suggested that snow persistence indices from Sentinel-2 images present a strong potential for the evaluation of distributed snow models in mountainous area. Hence, maps of the snow-covered area from the Copernicus Sentinel-2 satellite mission (Drusch et al., 2012) at 20-m resolution and at 5-day revisit time were considered as complementary

data to evaluate CHM simulations. Sentinel-2 images from 1st September 2017 to 31st August 2018 were processed using the snow retrieval algorithm that is currently used to produce the Theia snow collection (Gascoin et al., 2019). First, orthorectified top-of-atmosphere (level 1C) products were processed to bottom-of-atmosphere reflectances (level 2A) using the MAJA

software version 3.1 (Hagolle et al., 2017). MAJA output cloud mask and flat-surface reflectances were used as input to the LIS software version 1.5. The LIS algorithm is based on the Normalized Difference Snow Index (Dozier, 1989) and uses a

digital elevation model to better constrain the snow detection (Gascoin et al., 2019). Hansen et al. (2013) global forest product was used to mask out pixels with a tree cover density larger than 50% since the snow retrieval algorithm is not adapted to the detection of the snow cover in dense forest areas where the ground is obstructed by the canopy. To further avoid misclassifications due to forest obstruction or turbid water surfaces, the DEM was used to mask out pixels below 2000 m asl. The final snow product provided the following classification for each pixel: (i) no snow, (ii) snow, (iii) cloud, including cloud

shadows and (iv) no data.

Sentinel-2 snow cover maps at 20-m resolution were resampled to the same 50-m raster as the CHM output using a median filter. Maps of observed snow persistence (SP) indices at 50-m resolution were then derived following Macander et al. (2015) and Wayand et al. (2018). SP represents for each pixel the ratio between the number of snow-covered days divided by the total number of clear-sky observations (snow or no-snow). SP was computed using images from 1$^{st}$ April 2018 to 31$^{st}$ August 2018

and SP ranges from 0 (always snow-free) to 1 (always snow-covered). Over the study period, the mean number of clear-sky observations per pixel reached 18.6 days. The same calculation was carried out with CHM outputs to derive maps of simulated snow persistence indices. The same dates as the Sentinel-2 maps were used and for each date, the Sentinel-2 cloud and no-data masks were applied to make sure that the same pixels and dates were considered when computing observed and simulated SP indices. A grid cell was considered snow-covered if the snow thickness exceeded 5 cm (Gascoin et al., 2019). The agreement

between the simulated and the observed SP distributions was quantified as a function of elevation and slope orientation in the upper slopes for the three same sub-regions considered for snow depth (i.e., Kananaskis North, Kananaskis South, and Haig). Grid cells that were not covered by forest in the observations and in the simulations were considered for the analysis.

## 3. Results

The evaluation of the different wind downscaling methods is described in Sect. 3.1. The quality of the snowpack simulations

is then assessed in Sect. 3.2 using airborne LiDAR snow depth data and snow persistence indexes. A special emphasis is placed on the ability of the model to capture the elevation-snow depth relation as well as snow redistribution around wind-exposed ridges.

### 3.1 Wind field downscaling

Figure 3 compares the near-surface wind field obtained from a simple bilinear interpolation of the HRDPS wind field (Fig. 3a)

with the downscaled wind field obtained with (Fig. 3c) and without (Fig. 3b) the wind speed reduction in leeward areas. HRDPS provided a smooth wind field with relatively higher wind speeds in the northwestern part of the region characterized by high relief (Fig 3a) compared to the rest of the area. HRDPS did not reflect the local terrain information due to a horizontal resolution of 2.5 km. Combining the HRDPS wind field with precomputed microscale WindNinja simulations strongly altered





the near-surface wind field (Fig. 3b). The downscaled field contained the general pattern from the HRDPS modulated by the
local-scale terrain information added by WindNinja and reproduced some typical features of atmospheric flow in complex
terrain (e.g., Raderschall et al., 2008). In particular, the topography surrounding the main valleys channeled the downscaled
atmospheric flow, as illustrated by downscaled wind directions aligned parallel to the main valley axes. The presence of ridge
crests generated cross-ridge downscaled flow and associated crest wind speed-up. Downscaled wind speeds were the same on
the windward and leeward sides of crests, however, as expected with the mass-conserving version of Wind Ninja
(Wagenbrenner et al., 2016; 2019). For this reason, an additional downscaling step using the Winstral parameter to reduce the
wind speed in leeward areas was considered as described in Sect. 2.2.4 (Fig. 3c). Blue arrows on Fig. 3c correspond to leeward
areas sheltered from the atmospheric flow and characterized by low downscaled wind speed. This additional downscaling step
did not modify the wind direction in these areas.

Figure 4 gives the error metrics for the wind speed (Bias and RMSE) between the CHM simulations and observations at eight
automatic weather stations. The HRDPS without downscaling overestimated wind speed (positive bias) at all stations, except
the CNT station. This station is located on an exposed crest and presents the largest TPI value among the stations used for
model evaluation (Table 3). Downscaling wind to the CHM mesh using WindNinja microscale winds (experiment
HRDPS+WN) improved the error metrics (decrease in bias in absolute value and decrease in RMSE) at four of the stations
(BRP, HMW, FSR and CNT). In particular, the wind downscaling reduced the negative bias found in the HRDPS for the wind-
exposed CNT station, presumably because the downscaling captures ridge crest speed-up of wind velocity. Decreased model
performances were found at four neighbouring stations located around the Fortress Mountain Snow Laboratory, however
(CRG, FRG, FRS and FLG; Fig 3a). At these stations located along local ridges, the wind downscaling, accounting for crest
speed-up, increased the wind speed and led to a larger positive bias than the default HRDPS (Fig. 3b). Accounting for the
formation of zones of low wind speed in leeward areas in the downscaling method (experiment HRDPS+WN+Rc) was neutral
at two stations located at low elevation (BRP and HMW) and improved results at all remaining stations, except at CNT. Indeed,
a strong degradation of model performance was found at this station since it is placed on a sheltered triangle next to the crest
on the CHM mesh, leading to an unrealistic reduction of downscaled wind speed.

The wind downscaling method also modified the general wind direction (Figs. 3 and 5). Prevailing winds during the study
originated from the South (S; 180°) - South West (SW; 225°) at most of the stations whereas the HRDPS without downscaling
provided wind mainly from the SW - West (W; 270°). Improvements in wind direction when combining HRDPS and
WindNinja were found for about half of the meteorological stations. The large error at the CRG station illustrated that none of
the wind simulation considered in this study captured the complex features of the atmospheric flow around the Fortress
Mountain Snow Laboratory (Fig 3a).



## 3.2 Snowpack simulations

### 3.2.1 Observed and simulated snow distributions


To assess the ability of CHM to simulate small-scale features of snow accumulation and transport in alpine terrain, ALS-derived snow depths were compared with simulated snow depths for different CHM experiments for a sub-region of approximately 77 km$^2$ (Fig. 6). Observed snow depth was characterized by strong, spatial variability (Figure 6a). Shallow snow cover (generally less than 1 m) was found in the upper south- to northeast-facing slopes that were primary exposed to

wind (Fig. 5). Snow accumulated on the leeward side of these slopes (purple contours on Fig. 6a). Thick snow cover (> 4 m) existed at the bottom of steep slopes and in large concave cirques corresponding to avalanche deposition areas (red contours on Fig. 6a). The CHM simulation without lateral redistribution of snow (blowing snow and avalanching), *NoBS NoAv*, did not capture these features (Fig 6d). CHM without blowing snow and avalanche routines simulated a homogenous snow cover with reduced snow accumulation for some of the crest regions that are exposed to wind and prone to large surface snow sublimation.

A better visual agreement with observations was found when accounting for gravitational snow redistribution in CHM (Fig. 6c). In this configuration, CHM partially reproduced reduced snow accumulation on steep slopes and avalanche deposits were simulated at the bottom of these slopes (red contour on Fig. 6c). However, the model mostly underestimated the snow depth in these deposits compared to the observations (red contours on Fig. 6a) and did not capture the snow depth distribution in the upper slopes (purple regions on Fig. 6c). The reference CHM with lateral redistribution of snow and wind speed reduction in

leeward slopes, *BS Av Rc*, brought large improvements (Fig. 6b). Accounting for blowing snow redistribution reduced snow accumulation on windward slopes and locally increased snow deposition in the upper parts of leeward slopes (purple contours on Fig. 6b). It also led to a large increase in snow accumulation in avalanche deposition areas (red contours on Fig. 6b) that better corresponded with observed features of snow accumulation (Fig. 6a). However, *BS Av Rc* presented an overestimation of snow depth in some of the large valleys of the region (blue contours on Fig. 6) where avalanche deposition seemed to be

overestimated.

### 3.2.2 Elevation-dependency of snow depth

The agreement between observed and simulated snow depth distributions was examined as a function of elevation for three sub-regions (see Fig. 1) of the Kananaskis domain: Kananaskis North, Kananaskis South and Haig (Fig. 7 and 8). For each sub-region, the median of observed snow depth increased with elevation up to 2400 m followed by a decrease at the highest

elevations (Fig. 7), a relationship reported elsewhere (Grunewald et al., 2014; Kirchner et al., 2014). The same trend was found for the other percentiles shown on the whisker plots of the observed distributions of snow depth (Fig. 7). All CHM simulations overestimated the snow depth below 2100 m for each sub-region, partly explained by the tendency of HRDPS to overestimate precipitation at valley stations (see stations HMW and UPC on Fig. S1 in the supplementary material). The CHM simulation without lateral redistribution of snow (*NoBS NoAv*) did not capture the observed spatial variability within each elevation band

(Fig. 7). Instead, simulated average snow depth increased with elevation and diverged with observed decreased snow depth




recorded with the ALS survey. Therefore, the experiment *NoBS NoAv* presented an increase of the Wasserstein distance and RMSE with elevation (Fig. 8) associated with a continuous decrease in model performance with increasing elevation. Accounting for gravitational redistribution in CHM (experiment *NoBS Av*; orange boxes in Fig. 7) increased the spatial variability within each elevation band and reduced snow accumulation above 2400 m, especially for the Haig sub-region (Fig.

7c) characterized by steep slopes prone to avalanching (Fig. 1b). The experiment *NoBS Av* led to improved Wasserstein distance at all elevation for each sub-region compared to the experiment *NoBS NoAv* (Fig. 8). Snow depth above 2300 m for all sub-regions was still overestimated, however (Fig. 7). The increase in RSME below 2400 m (Fig. 8) suggested that experiment *NoBS Av* did not capture the location of avalanche deposits well.

Including blowing snow redistribution strongly affected model results. As expected, it increased the spatial variability of

simulated snow depth within each elevation band compared to experiments *NoBS NoAv* and *NoBS Av* (Fig. 7). When the wind speed reduction in leeward areas was not simulated (experiment *BS Av NoRc*), CHM underestimated the median snow depth (as well as the 1st and 3rd quartile) above 2500 m compared to observations. This underestimation increased with elevation and was largest for the elevation band 2900-3100 m. Including the recirculation effect when simulating blowing snow (experiment *BS Av Rc*) strongly improved the ability of the model to capture the distribution of snow depth at high-elevation (above 2700

m for Kananaskis North, Fig. 8a; above 2500 m for Kananaskis South, Fig. 8b and above 2100 m for Haig, Fig. 8c). Overall, the experiment *BS Av Rc* captured the observed shape of the elevation-snow depth relation for each sub-region (Fig. 7). Below 2300 m, experiments *BS Av NoRc* and *BS Av Rc* overestimated the value of the 95th percentile of the snow depth distribution compared to observations (Fig. 7). These two configurations of CHM also led to a larger Wasserstein distance (between 1900 m and 2100 m) and larger RMSE (below 2500 m) compared to experiments *NoBS NoAv* and *NoBS*  In short, these results

quantify the degree of bias of gravitational redistribution to lower elevation and the erroneous location of avalanche deposits observed on Fig. 6b.

### 3.2.3 Snow distribution around ridges

The observed and simulated snow depth distributions were compared for the upper slopes of the domain (defined in Sect. 2.3.2), particularly exposed to wind-induced snow transport (Fig. 9). The CHM simulation without lateral redistribution of

snow, *NoBS NoAv,* presented a systematic overestimation of snow depth for all slope orientations (Fig. 9, Top) and yielded the worst Wasserstein distance metric among all simulations (Fig. 9, Bottom). Including gravitational redistribution reduced the positive bias and the Wasserstein distance. This reduction was not found for some slope orientations, however (W, SW and S orientations for Kananaskis North, Fig. 9a, and SW and S orientations for Kananaskis South, Fig 9b). The moderate values of the slope angle generally found for these orientations were not sufficient to trigger gravitational snow redistribution in

SnowSlide. For example, the percentage of slope values larger than 40° is only 9% for the SW and S orientations for Kananaskis North, compared to 43% and 63% for the N and NE orientations for the same region, respectively. Accounting for blowing snow redistribution without wind speed reduction in leeward areas generated a systematic underestimation of snow depth for all slope orientations and sub-regions (Fig. 9, Top). This negative bias in snow depth indicates that snow erosion on the



windward slopes (S to NW orientations) was overestimated for experiment *BS Av NoRc*. Thinner-than-observed snow depth

on the upper part of the leeward slopes (N to SE orientations on Fig. 9, Top) suggests that the wind-blown snow eroded in excess from the windward slopes was transported by PBSM-3D to the lower part of the leeward slopes due to the absence of wind recirculation in the driving wind field used by experiment *BS Av NoRc* (Fig. 3b). A similar underestimation of snow depth on the windward slopes exists for experiment *BS Av Rc*. Due to the activation of the wind speed reduction in leeward areas, however, this experiment simulated snow deposition in the upper part of these areas. The experiment *BS Av NoRc* led

to an overestimation of snow depth on the leeward slopes for Kananaskis North and South (N to SE orientations, Fig. 9a b) and nearly unbiased estimation of snow depth for Haig (NE to SE orientations, Fig. 9c). Overall, experiment *BS Av Rc* provided the best performances in term of Wasserstein distance for the windward slopes of all sub-regions (Fig 9. Bottom) despite the negative bias in snow depth for these areas. Performances were more mixed for leeward slopes: Haig showed an improvement compared experiment *NoBS Av* (Fig. 9f) whereas worst performances were obtained for Kananaskis North (Fig. 9d).

**3.2.4 Snow Persistence**

Figure 10 shows the maps of observed snow persistence indexes as well as the indexes derived from two CHM simulations. Observed SP (Fig. 10a) presented similar patterns compared to the observed distribution of snow depth in late April (Fig 6a), showing that snow persistence patterns are primarily controlled by the patterns of peak snow accumulation (Wayand et al., 2018). Avalanche deposits identified on Fig. 6a corresponded to maximal SP values whereas low SP values were found near

ridges lines, exposed to wind. Overall, the Pearson correlation coefficient between observed snow depth and SP reached 0.69, 0.68 and 0.75 for Kananaskis North, Kananaskis South and Haig, respectively. Without accounting for lateral snow redistribution (experiment *NoBS NoAv*), CHM-simulation derived SP values were dependent upon the elevation and slope orientation (Fig. 10c), primarily due to the impact of solar radiation on simulated snow ablation. Without lateral snow redistribution, snow accumulation was spatially uniform (Fig. 6d and 7). Lower values of simulation derived SP were found

in the lower south-facing slopes whereas steep slopes on northern faces had SP values close to 1.0. The simulation derived SP was strongly modified with experiment *BS Av Rc* (Fig. 10b), similarly to the effect found for snow depth near peak snow accumulation (Fig. 6b). In this experiment, windward slopes systematically presented low SP values and maximal SP values close to 1.0 were found at the bottom of slopes due to gravitational redistribution of snow that prevented from snow persistence on steep slopes.

Figure 11 shows how accurately the different CHM experiments were able to reproduce the observed SP distributions as a function of elevation. The simulation derived and observed SP distributions are shown on Fig. S2. Model performances for snow persistence were generally in agreement with those for snow depth presented at Fig. 8. Experiments without blowing snow redistribution (*NoBs NoAv* and *NoBs Av*) overestimated snow persistence at all elevations with a positive bias increasing with elevation for experiment *NoBs NoAv*. Including blowing snow redistribution in experiments *Bs Av NoRc* and *Bs Av Rc*

significantly decreased snow persistence, mainly above 2300 m. The absence of wind speed reduction in leeward areas (experiment *Bs Av NoRc*) led to negative bias of snow persistence above 2700 m and a decrease in model performances



compared to experiment *Bs Av Rc* that includes recirculation effects. Below 2500 m, experiments *Bs Av NoRc* provided the best performances in terms of bias and Wasserstein distance. Consistent results compared to snow depth were also obtained when considering how observed and simulated SP distributions vary with slope orientation in upper slopes (Fig. 12). For

example, the tendency of experiment *Bs Av Rc* to overestimate snow accumulation on the leeward slopes of Kananaskis North led to a clear overestimation of snow persistence on these slopes (Fig. 12a) and a degradation of model performance compared to a simulation without blowing snow redistribution (Fig. 12d).

## 4. Discussions

### 4.1 Modelling of mountain snowpack

This study presents a new high-resolution modelling strategy for mountain snowpack combining atmospheric forcing from a NWP system at convection-permitting scale with the multi-scale, snowdrift-permitting model CHM. Several CHM configurations were tested to highlight how missing physical processes influenced the performances of snowpack simulations at snowdrift-permitting scales (50 m in this study). Lateral snow redistribution (blowing snow and avalanching) were required to capture natural variations in snow depth and its persistence, a finding that accords with Winstral et al. (2013) and Hanzer et

al. (2016). These results differed from Revuelto et al. (2018) who showed that a distributed snowpack scheme without lateral snow redistribution can provide accurate estimation of snow cover variability. This discrepancy may arise from (i) the resolution of 250 m used in Revuelto et al. (2018) for which lateral redistribution processes are partially sub-grid and (ii) the absence of ALS data and high-resolution satellite images to evaluate their snowpack simulations. Accounting for gravitational redistribution reduced snow accumulation and persistence in steep slopes in agreement with the findings of Bernhardt and

Schulz (2010). This was not sufficient and snow depth and persistence were still overestimated for the upper slopes exposed to wind. The CHM simulation (*Bs Av Rc*) that included blowing snow redistribution and avalanching was required to capture the decrease in snow depth at high-elevation (above 2500 m); it also improved the elevation-snow depth relationship for all sub-domains of the Kananaskis domain. Similar elevation-snow depth relationships presenting a snow depth maximum below the highest elevations have also been reported for other mountainous regions (Grunewald et al., 2014; Kirchner et al., 2014).

Our results suggest that accounting for blowing snow redistribution and avalanching in distributed snowpack simulations is crucial to accurately simulating the elevation-snow depth relationships in high mountain terrain.

Results of blowing snow redistribution simulations in CHM were sensitive to the quality of the driving wind field, at the mountain range scale (> 100 km$^2$). This observation builds on the similar findings of Mott and Lehning (2010) and Musselman et al. (2015) based on ridge-scale simulations of snow depth. High-resolution wind fields obtained using the mass-conversing

version of the diagnostic wind model WindNinja (Forthofer et al., 2014) presented some features of atmospheric flow in alpine terrain (e.g. valley channeling, crest speed-up) but they did not capture the formation of recirculation areas on leeward slopes. This lack of recirculation led to lower-than-observed snow deposition and persistence on leeward slopes. These results highlight the limitations of mass-conversing diagnostic wind models for blowing snow modelling in alpine terrain. Combining





high-resolution wind fields from WindNinja with a terrain-based parameter (Winstral and Marks, 2002) allowed identifying
potential areas of flow separation on leeward slopes and improved simulations of the elevation-snow depth relationship and of
the snow distribution and persistence around ridges. This simulation was still impacted by an overestimation of snow erosion
on windward slopes and subsequent deposition on leeward slopes, likely arising from uncertainties associated with the wind
downscaling method and limitations in CHM parameterizations discussed in Sect. 4.3 and 4.4.

## 4.2 Importance of high-resolution distributed evaluation data

The evaluation of the wind downscaling methods versus point measurements did not show systematic improvements compared
to the original HRDPS wind field, consistent with studies of high-resolution wind modelling in complex terrain (e.g., Horvath
et al., 2012; Vionnet et al., 2015). Model results in Sect. 3.1 highlight the challenge of evaluating wind simulations at locations
near peaks or ridges due to approximation in the location of the stations as previously mentioned in Fiddes and Gruber (2014)
and Winstral et al. (2017). On the other hand, differences between the wind downscaling methods were clearly identified and
quantified when evaluating the snow simulations using distributed data. ALS snow depth and snow persistence indexes derived
from Sentinel-2 allowed for targeted model evaluation in area of interest such as the upper slopes exposed to wind-induced
snow transport. These results confirm the large potential of ALS snow depth data for detailed model evaluation (e.g., Hanzer
et al., 2016; Hedrick et al., 2018). In addition, they show that snow persistence indexes derived from freely available Sentinel-
2 images (Wayand et al., 2018) can generally support similar conclusions than those derived from ALS snow depth. This
highlights that these indexes can be used to evaluate large-scale snowpack simulations at snowpack-permitting scales in regions
that are not covered by LiDAR.

Two types of metrics were used when using ALS snow depth data for model evaluation: RMSE and Wasserstein distance.
RMSE corresponds to a traditional "point-to-point" verification metric. Such metric may favor homogenous snow cover
simulations. Indeed, a snow cover simulation including avalanching may present a degree of realism but errors in the exact
location of the avalanche deposits may increase RMSE compared to a simulation without avalanching due to the double-
penalty problem (e.g., Nurmi, 2003). This issue is often encountered when evaluating the ability of high-resolution atmospheric
model to simulate localized events such as convective precipitation (e.g., Clark et al., 2016). The Wasserstein distance
(Rüschendorf, 1985) was used in this study as a complementary metric to evaluate the agreement between observed and
simulated distributions for specific areas (elevation bands or specific slope orientations). This metric may lead to a perfect
match even if the observations and the simulations are not co-located, however. This highlights the need to consider several
verification metrics with identified strengths and limitations. In the future, more advanced verifications methods such as the
neighborhood methods developed in the atmospheric community (Ebert et al., 2013) could be considered.

## 4.3 Uncertainties in the atmospheric driving data

This study used a wind downscaling method inspired by Barcons et al. (2018) and developed for large areas. Part of the
uncertainty associated with this method comes from the value of the radius of influence used to compute the transfer functions



(Sect. 2.2.4). A value of 1 km was selected for this study, similar to Barcons et al. (2018) as the resolutions of the mesoscale atmospheric models were similar between the two studies (2.5 km in this study and 3 km in Barcons et al., 2018). Further work is required to adapt this value to the resolution of the mesoscale atmospheric models and to the terrain complexity. The accuracy of the wind downscaling was also influenced by the quality of the diagnostic wind model used to generate microscale wind

fields. In particular, the mass-conserving version of WindNinja used in this study failed to simulate the formation of recirculation areas on leeward slopes that are one of the main features of atmospheric flow in alpine terrain (Raderschall et al., 2008; Gerber et al., 2017). The practical method relying on the Winstral parameter (Winstral and Marks, 2002) proposed to overcome this limitation is affected by strong assumptions on the value of the critical angle for flow separation (Wood, 1995) and of the transfer function in recirculation zones. Gerber et al. (2017) showed that atmospheric stability affects the value of

this angle and the development of leeside recirculation. In addition, the wind direction is not modified in these areas, contrary to simulations resulting from CFD models or atmospheric models in LES mode (Gauer, 1998; Mott and Lehning, 2010; Vionnet et al., 2017). Improvements in the wind downscaling could be achieved using such models to generate the library of wind fields, as proposed by Barcons et al. (2018). Different conditions of atmospheric stability could also be considered (e.g., Gerber et al., 2017). Final selection of the wind downscaling strategy will ultimately be a trade-off between model complexity,

accuracy and computational costs and will vary as a function of model applications.

All the atmospheric driving data for CHM were obtained from the HRDPS, the Canadian NWP system using GEM at 2.5 km (Milbrandt et al., 2016). It consisted of successive short-range forecasts combined to generate a continuous atmospheric forcing. Such approach has been used previously to generate atmospheric forcing for snowpack models in mountainous terrain (Horton and Jamieson, 2016; Quéno et al., 2016; Vionnet et al., 2016; Luijting et al., 2018). Error in the snowpack variables

can grow with time due to errors in the successive forecasts, especially those due to precipitation biases (e.g., Vionnet et al., 2019). Errors in the longwave and shortwave radiative input can also significantly affect snowpack simulations (Lapo et al., 2015; Quéno et al., 2020). The downscaling techniques used to adapt the HRDPS forcing to the CHM mesh likewise contribute to the uncertainty in the quality of the meteorological input. Monthly fixed altitudinal gradients were used to adjust the near-surface temperature and humidity forcing; this method might be further improved using upper-air HRDPS temperatures and

humidity fields (e.g., Jarosch et al., 2010; Fiddes et al., 2014). Contributions from the surrounding topography to longwave irradiance were also neglected, despite its impact on the snowpack energy balance on inclined slopes (Pluss and Ohmura, 1997). Ensemble simulations based on ensemble meteorological forcing (e.g., Vernay et al., 2015) and ensemble downscaling methods (Marsh et al., 2020a) could be used to investigate the impact of these sources of uncertainties.

### 4.4 Limitations in the physical parameterizations in CHM

The model evaluation for the upper slopes exposed to wind showed that CHM simulations including blowing snow tend to overestimate snow-redistribution across slopes subject to wind erosion and deposition. These results were obtained for a CHM mesh with a typical area of 50 m × 50 m near the crest lines. Mott and Lehning (2010) found a similar overestimation of snow-redistribution in simulations of the snow cover evolution for a crest of the Swiss Alps using the Alpine 3D model (Lehning et



al., 2008) running at 50-m grid spacing. They showed that increasing the model resolution finer than 10 m increased snow
accumulation on windward side due a more accurate representation of small-scale terrain features trapping snow on the windward side. These results suggest that the absence of any subgrid topography effects on snow transport in CHM can partially explain the overestimation of snow redistribution from windward slopes to leeward slopes and subsequent avalanching. In addition, CHM uses the formulation for the threshold friction velocity for snow transport of Li and Pomeroy (1997) that only depends on snow presence and air temperature. Such parameterization does not account for the effect of snow
fragmentation during blowing snow events (Comola et al., 2017) which may lead to an underestimation of the threshold friction velocity and an overestimation of blowing snow occurrence in alpine terrain (Vionnet et al., 2013). Finally, CHM cannot simulate the formation of snow plumes at crest lines due to the representation of the suspension layer in PBSM-3D (Marsh et al., 2020a). Mass loss due to the advection of blown snow particles in upper atmospheric layer and subsequent sublimation may thus be underestimated by CHM. Such a limitation is also found for more advanced blowing snow schemes (see for
example Fig. 6 of Groot Zwaaftink et al., 2011 and Fig. 8.6 of Vionnet, 2012).

Gravitational snow redistribution is simulated in CHM with the SnowSlide scheme (Bernard and Schulz, 2010). Model results showed that CHM can reproduce the formation of snow accumulations due to avalanching that visibly correspond with the observations. However, the increase in RMSE for snow depth at low elevation for all simulations including avalanching suggests that CHM does not effectively capture the true location of these deposits. SnowSlide relies on a maximum holding
capacity of snow that only depends on the slope angle and does not consider the small-scale terrain roughness, limiting the ability of the scheme to reproduce snow accumulation for steep faces (Sommer et al., 2015). In addition, the exact location of avalanche deposits is influenced by avalanche dynamics (Pudasaini and Hutter, 2007) which are not reproduced in SnowSlide. CHM also does not represent snowfall enhancement due to interactions between the flow field and the local cloud formation as well as the preferential deposition of snowfall resulting from pure particle flow interaction (Lehning et al., 2008; Vionnet
et al., 2017; Mott et al., 2018). Gerber et al. (2019) suggested that, when combined, these two effects can increase snow accumulation on the leeward side of mountain ridges by 26-28%. In the current version of CHM, wind-induced snow transport is the only process responsible for additional snow deposition on leeward slopes. A study is in progress in the Canadian Rockies to better assess the impact of terrain–flow–precipitation interactions on snow accumulation in the region. Finally, uncertainties associated with the Snobal snowpack scheme were not quantified in this study. In particular, errors in simulated snow density
can affect the comparison between observed and simulated snow depth (Raleigh and Small, 2017; Lv and Pomeroy, 2020), despite the use of an improved snow density algorithm for Snobal (Hedrick et al., 2018). Inaccurate estimations of the ground heat flux may also affect the simulation of the snow cover duration (Slater et al., 2017). Pritchard et al. (2020) showed how multi-physics ensemble snow modelling can be applied to assess uncertainties on distributed snowpack simulations and a similar framework could be applied to CHM, including uncertainties in PBSM-3D and SnowSlide.





## 5. Conclusions


This study presents a new multi-scale modeling strategy of mountain snowpack over large regions. It combines (i) atmospheric forcing from the Canadian GEM NWP system at a convective-permitting scale (Milbrandt et al., 2016), (ii) a meteorological downscaling module including a wind downscaling strategy relying on the diagnostic wind model WindNinja (Forthofer et al., 2014) and (iii) the multi-scale snowdrift-permitting model CHM (Marsh et al., 2020a, b). This system was used to simulate

the snowpack evolution for an entire snow season over a domain of 958 km² in the Kananaskis Valley of the Canadian Rockies. Wind simulations were evaluated using data from automatic stations in the domain. The distributed evaluation data for the snowpack simulations consisted of maps of snow depth derived from airborne LiDAR and snow persistence indexes derived from optical satellite imagery. Several configurations of CHM were tested to assess the effect of the wind field downscaling, and the impact of process representation on snowpack simulations at snowdrift-permitting scales.

The main conclusions of this study are that:

- Pre-computed wind fields at 50-m grid spacing with the WindNinja model can be combined efficiently with output of the Canadian NWP system at 2.5 km grid spacing to produce hourly driving wind fields including small-scale topographic features. The mass conserving version of WindNinja used in this study cannot reproduce leeside flow recirculation, however. The Winstral terrain parameter, $Sx$, provides a solution to identify potential recirculation areas

and adjust accordingly the wind field downscaled with WindNinja.

- Snowpack simulation without lateral snow redistribution (blowing snow and gravitational snow redistribution) cannot capture the spatial variability of snow cover in alpine terrain and overestimates snow depth and snow cover duration at high elevations. Including gravitational redistribution improved model results and reduced snow depth at high elevations. Snow depth and snow cover duration was still overestimated around ridge lines exposed to winds.

- Snowpack simulation including blowing snow and gravitational snow redistribution provided the best estimates of the shape of the elevation-snow depth relation across the Kananaskis region and reproduced the decrease in mean snow depth found at high elevation. These results were obtained for a CHM experiment driven by a wind field including the wind speed reduction in leeward areas. Removing these zones led to a systematic underestimation of snow depth around ridges, partially due to an underestimation of snow deposition on leeward slopes. These results

highlight that wind fields without lee-side slowdown are not sufficient to simulate snow redistribution in mountainous terrain.

- Snowpack simulations including blowing snow and gravitational snow redistribution overestimated snow redistribution from windward to leeward slopes and subsequent avalanching. This is potentially due to the absence of subgrid topographic effects on snow transport in CHM.

- High-resolution snow persistence indexes derived from Sentinal-2 presents a strong potential for the detailed evaluation of distributed snowpack models, in particular in regions where Airborne LiDAR snow depth data are not
available. These indices can be used for model evaluation targeting specific areas (e.g. ridges lines exposed to intense wind-induced snow redistribution, avalanche deposition areas).

The results of this study demonstrate that CHM at snowdrift-permitting scale constitutes a promising tool for large-scale modelling of mountain snowpack. Future work will combine (i) improvements in the physical parameterizations in CHM and in the driving wind fields, (ii) large scale simulations across the Western Canadian Cordillera, and (iii) improvements of CHM simulations with assimilation of high-resolution observations.

**Code availability**

The open-source CHM model code (Marsh et al., 2020b) is available at https://github.com/Chrismarsh/CHM. The *mesher*
algorithm (Marsh et al., 2018) is available at https://github.com/Chrismarsh/mesher. The high-resolution wind library has been generated using the WindNinja diagnostic wind model (Forthofer et al., 2014; https://weather.firelab.org/windninja/) and the Windmapper tool (https://github.com/VVionnet/Windmapper). The Sentinel-2 snow cover maps were generated from level 1C images using the free software MAJA (https://logiciels.cnes.fr/en/content/maja) and the open source software LIS (https://gitlab.orfeo-toolbox.org/remote_modules/let-it-snow/).

**Data availability**

CRHO meteorological and snow observations are available through the web portal http://giws.usask.ca/meta/. HRDPS forecasts are distributed on the Canadian Surface Prediction Archive (CaSPAr; https://caspar-data.ca/). Sentinel-2 level 1C data were obtained from the Plateforme d'Exploitation des Produits Sentinel (https://peps.cnes.fr). Final snow cover maps over the Kananaskis Valley are availabe on Zenodo (Gascoin, 2020). The HRDPS forcing file and the ALS data used in this study
are available on request to the authors.

**Author contribution**

VV, CM, BM and JP designed the study and the modelling strategy. VV and CM developed the wind downscaling module in CHM. VV was responsible for the analysis of the results and the preparation of the manuscript. BM, JS, and KM processed and provided the Airborne-LiDAR snow depth data. SG processed and provided the Sentinel2 snow cover images. NW
provided a pre- and post-processing toolkit for CHM. All authors contributed to the preparation of the manuscript.

**Competing interest**

The authors declare that they have no conflict of interest.



**Acknowledgements**

This study was supported by the Global Water Future programme funded by the Canada First Research Excellence Fund.
Support from the Canadian Foundation for Innovation, Canada Research Chairs Program, NSERC and Tula Foundation is gratefully acknowledged. Martyn Clark (USask) and Barbara Casati (ECCC) are thanked for their helpful scientific discussions. Special thanks to Logan Fang (USask) and Greg Galloway (USask) for providing quality-controlled snow and meteorological data. SG acknowledges the Centre National d'Etudes Spatiales (CNES) for granting him access to its high performance computer.

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



**Table 1: Characteristics of the mesh used in this study. The vertical error corresponds the root mean square error to the underlying reference topographic raster.**

| Name of the mesh | Minimum triangle area (m$^2$) | Maximum triangle area (m$^2$) | Median triangle area (m$^2$) | Vertical error (m) | Number of triangles |
|---|---|---|---|---|---|
| $M^{15}_{50}$ | $50 \times 50$ | $250 \times 250$ | $63 \times 63$ | 15 | 101 700 |


**Table 2: CHM simulations (experiments) used in this study. *Rc* indicates CHM simulations using wind fields from the downscaling method accounting for wind speed reduction in leeward areas. HRDPS refers to the High Resolution Deterministic Prediction System and WN to Wind Ninja. See text for more details.**

| Name | Driving Wind Field | Gravitational redistribution | Blowing Snow |
|---|---|---|---|
| *NoDown* | HRDPS | No | No |
| *NoBS NoAv* | HRDPS + WN + Rc | No | No |
| *NoBS Av* | HRDPS + WN + Rc | Yes | No |
| *BS Av NoRc* | HRDPS + WN | Yes | Yes |
| *BS Av Rc* | HRDPS + WN+ Rc | Yes | Yes |

**Table 3: Meteorological stations used for wind evaluation. TPI refers to the Topographic Position Index and is defined as the difference between the elevation of the station minus the mean elevation within a 2-km radius from this station. The location of the stations is shown on Fig. 1.**

| Full Name | Code | Latitude (°) | Longitude (°) | Elevation (m) | TPI (m) |
|---|---|---|---|---|---|
| Centennial Ridge | CNT | 50.9447 | -115.937 | 2470 | 248 |
| Fisera Ridge | FSR | 50.9568 | -115.2044 | 2325 | -10 |
| Hay Meadow | HMW | 50.9441 | -115.1389 | 1492 | -33 |
| Fortress Ledge | FLG | 50.8300 | -115.2285 | 2565 | 216 |
| Fortress Ridge | FRG | 50.8364 | -115.2209 | 2327 | 99 |
| Fortress Ridge South | FRS | 50.8382 | -115.2158 | 2306 | 129 |
| Canadian Ridge | CRG | 50.8215 | -115.2063 | 2211 | 68 |
| Burtsall Pass | BRP | 50.7606 | -115.3671 | 2260 | -90 |


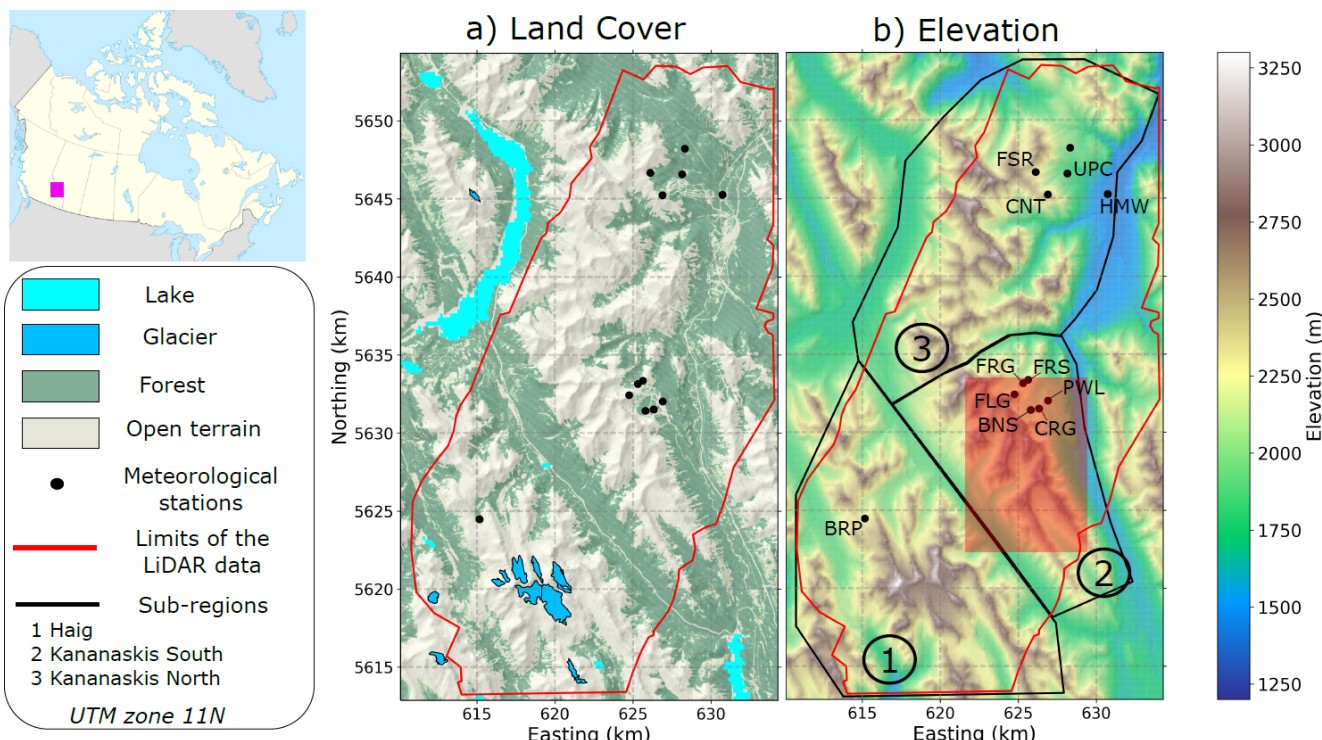

**Figure 1: a) Land cover map and b) elevation map of the Kananaskis Valley, Alberta, Canada study domain. The glacier mask is taken from the Randolph Glacier Inventory version 6.0 (Pfeffer et al., 2014). The red-shaded area correspond to the area shown in Fig. 2, 3 6 and 10. The characteristics of the meteorological stations are given in Tab. 3. Areas labelled from 1 to 3 correspond to sub-regions used in the analysis of the results (see Sect. 2.3.2).**


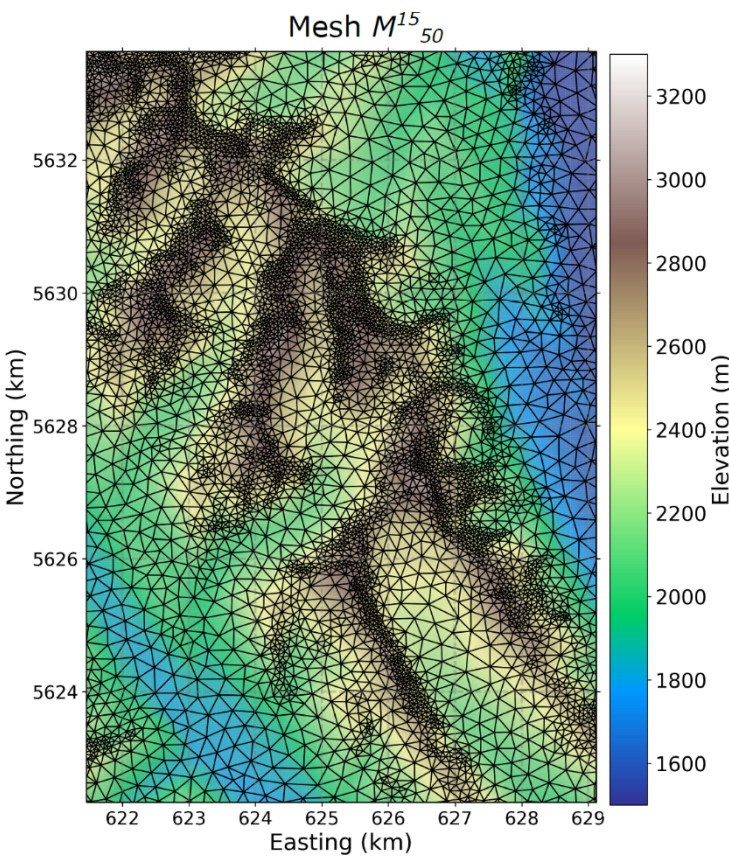

**Figure 2: Variable resolution triangular mesh used in this study over a sub-area of the Kananaskis domain. The location of this sub-area corresponds to the red-shaded area shown on Fig. 1b. The underline DEM was taken from the SRTM mission at 1 arc second.**
**Figure 3: Near-surface wind field on 10 September 2017 at 18 UTC from (a) HRDPS without downscaling, (b) HRDPS downscaled to the CHM mesh $M^{15}_{50}$ using WindNinja and (c) same as (b) but including a parameterization for the formation of recirculation zones on leeward slopes (see Sect. 2.2.4 for more details). The location of this sub-area corresponds to the red-shaded area shown on Fig. 1b. Arrows indicate wind direction while colours indicate wind speed. One arrow is shown every 250 m for clarity. The underlying topography is shown using hill shading. Effects of vegetation on the simulated wind fields are not shown in these maps.**





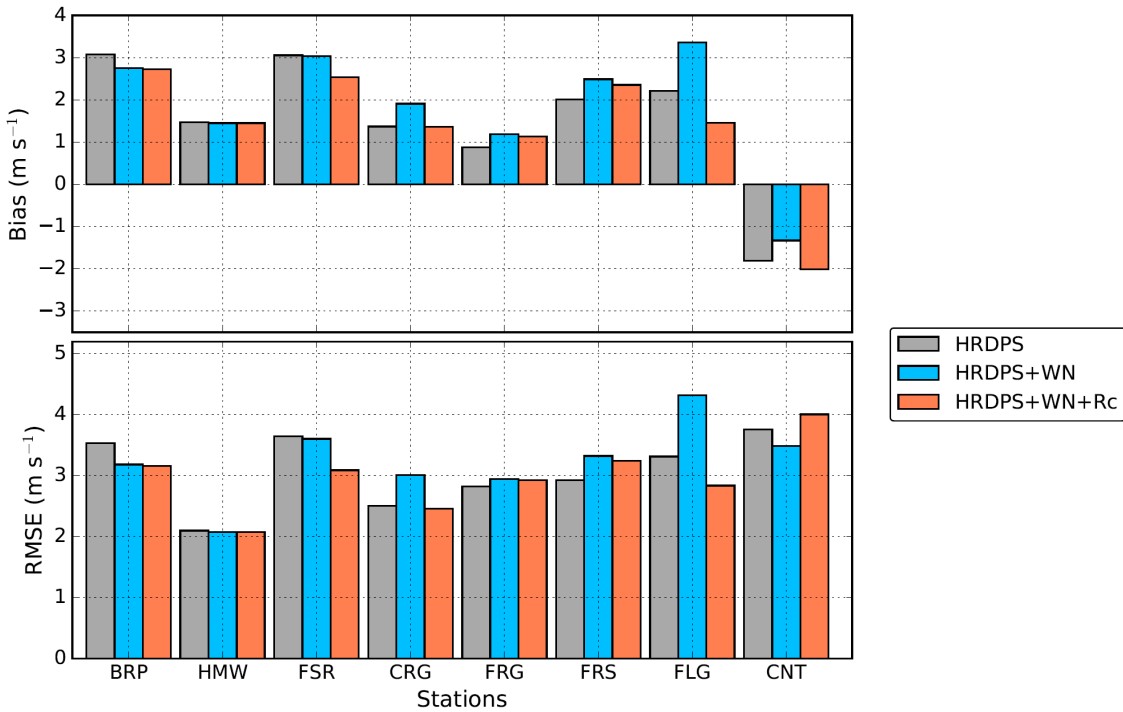


**Figure 4: Evaluation of simulated wind speed using different downscaling methods: (Top) Bias and (Bottom) Root Mean Square Error (RMSE). Grey colours show the HRDPS wind speed without downscaling, blue colours show the HRDPS wind speed combined with WindNinja microscale winds (HRDPS+WN) and red colours show the same configuration as HRDPS+WN including in addition the wind speed reduction on leeward slopes. Stations used for evaluation are classified by increasing TPI (Table 3). Their location is shown on Fig. 1.**



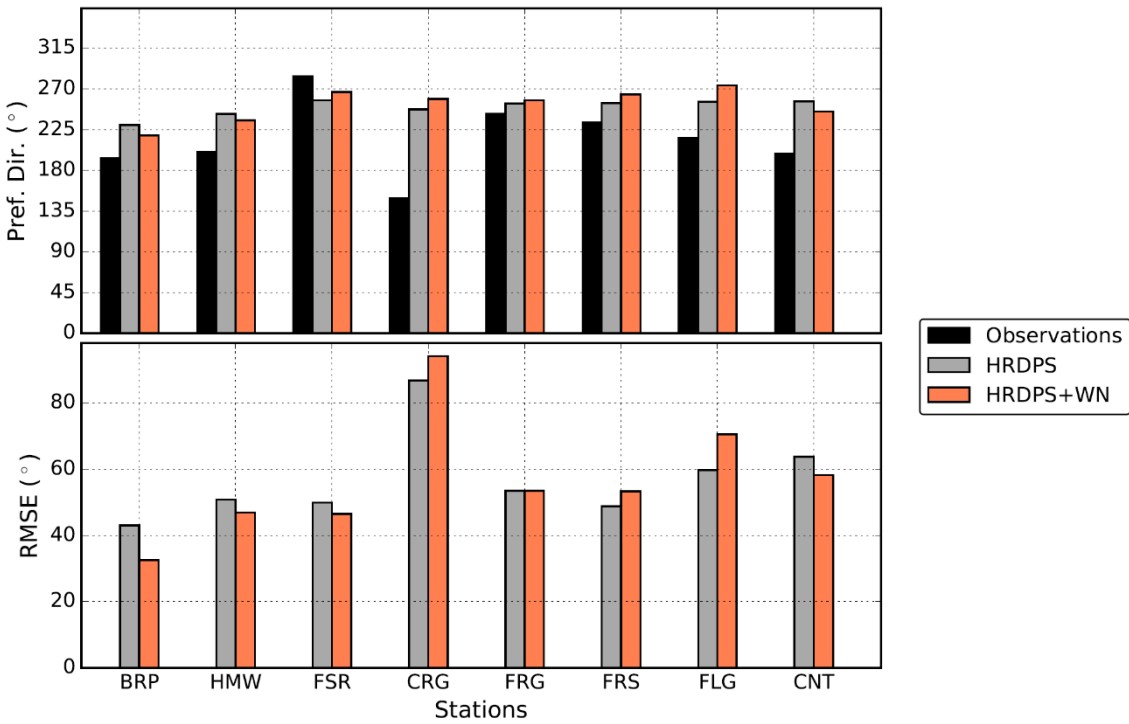

**Figure 5: Same as Fig. 4 for wind direction: (Top) Preferential wind direction and (Bottom) Root Mean Square Error (RMSE). Error metrics were computed for wind direction only when observed wind speed was larger than 3 m s⁻¹. Configuration HRDPS+WN+Rc is not shown since the wind direction is unchanged compared to HRDPS+WN.**






**Figure 6: Snow depth on 27 April 2018 (a) measured by ALS and simulated by three CHM configurations: (b) *BS Av Rc*, (c) *NoBS Av*, and (d) *NoBS NoAv* (Table 3). Properties of snow depth distribution in areas with coloured contours are discussed in the text. Pixels covered by tall vegetation in the observations and in the simulations are excluded from the comparison and appear in grey. Black isolines correspond to Δz = 50 m and the location of this region in shown on Fig. 1b.**






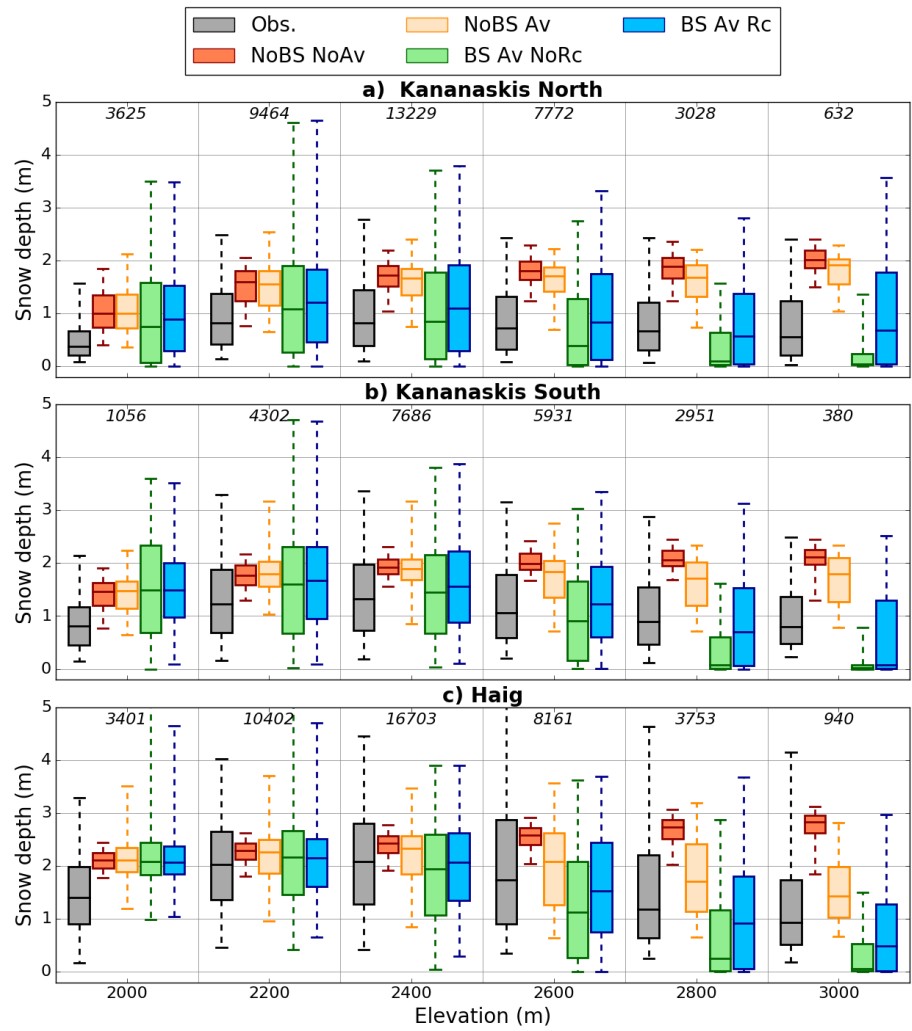

**Figure 7: Boxplots showing the distributions of observed and simulated snow depth per 200-m elevation bands for three sub-regions. The location of these sub-regions is shown on Fig. 1b. Results of four CHM experiments are shown. The numbers in italic indicate the number of grid points within each elevation band. The whiskers show the 5th and 95th percentiles and outliers are not plotted.**


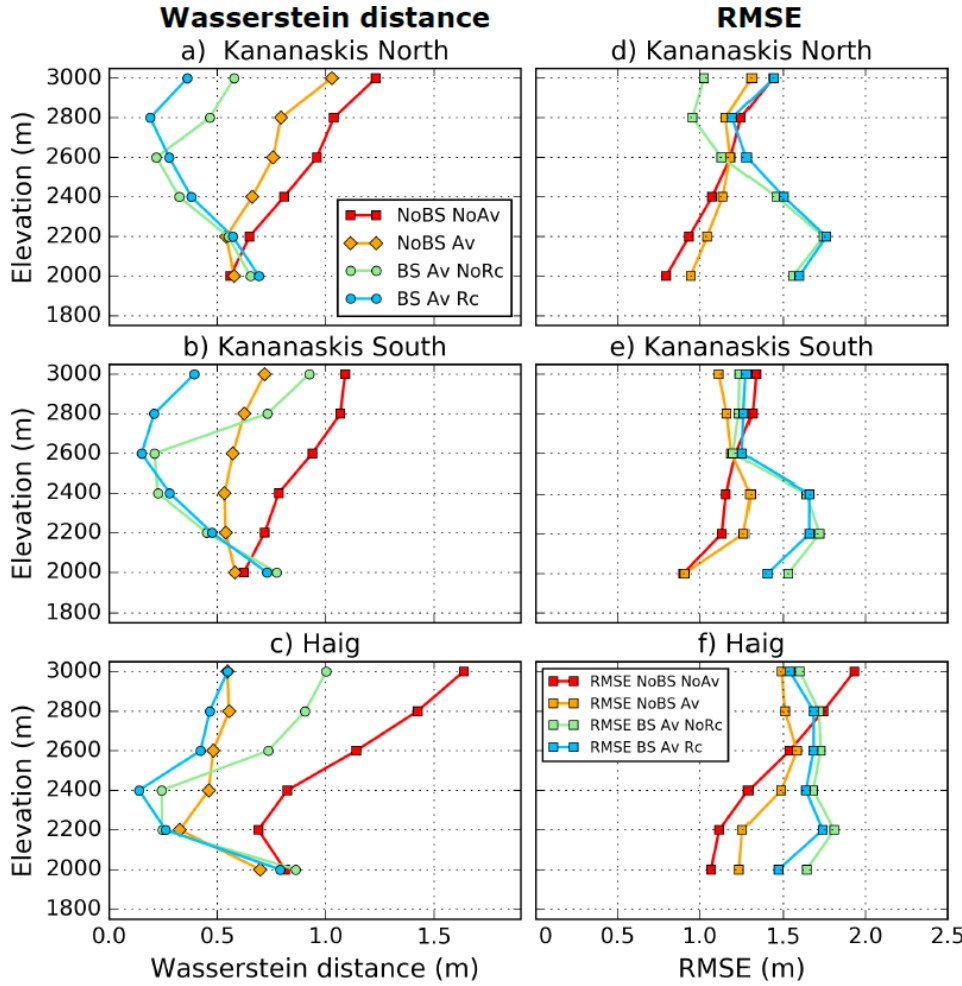

**Figure 8: Wasserstein distance and RMSE between observed and simulated snow depth distribution as a function of elevation for four CHM experiments and three sub-regions. The location of these sub-regions is shown on Fig. 1.**

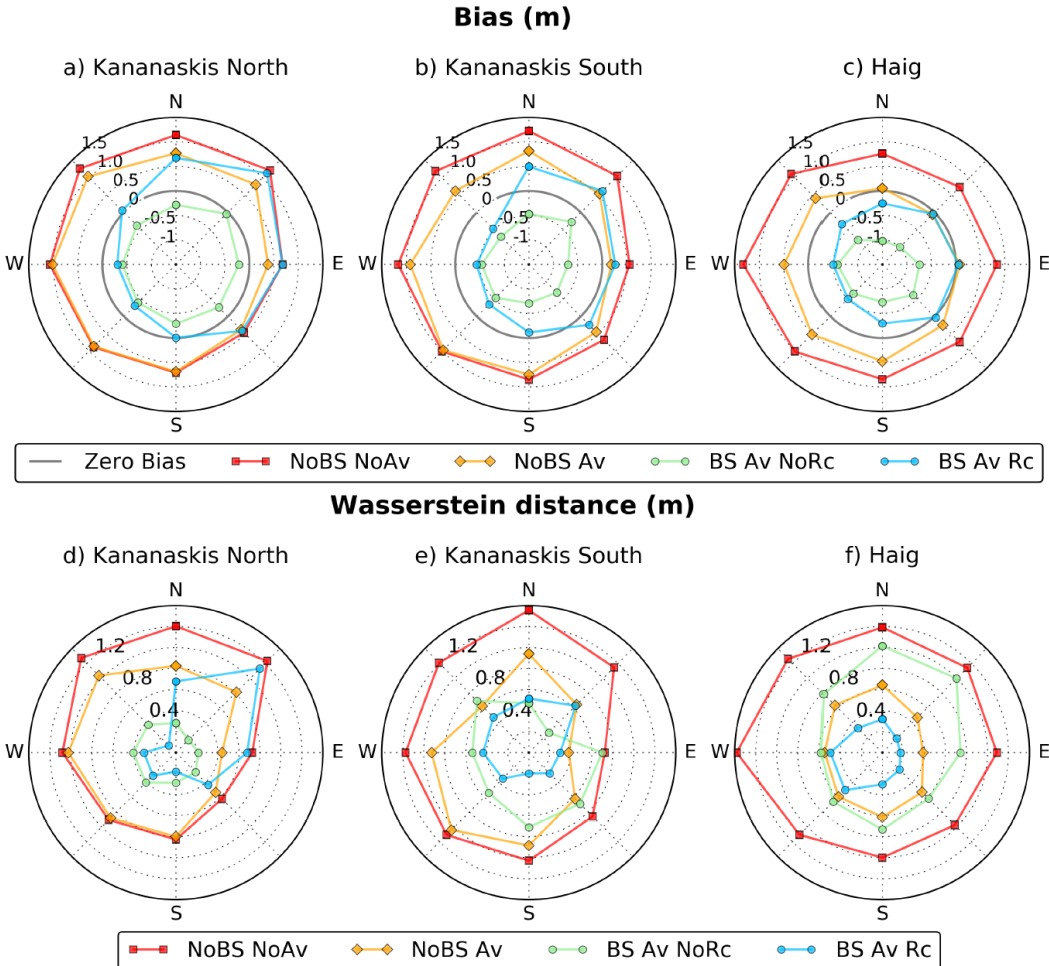

Figure 9: Bias (Top) and Wasserstein distance (Bottom) between observed and simulated snow depth distribution in upper slopes as a function of slope orientation for four CHM experiments and three sub-regions. The location of these sub-regions is shown on Fig. 1. Upper slopes are defined as regions of TPI larger than 150 m (see Sect. 2.3.2). The thick grey circles on graphs a, b and c indicate a zero bias. Values outside this circle indicate a positive bias whereas values within this circle indicate a negative bias.
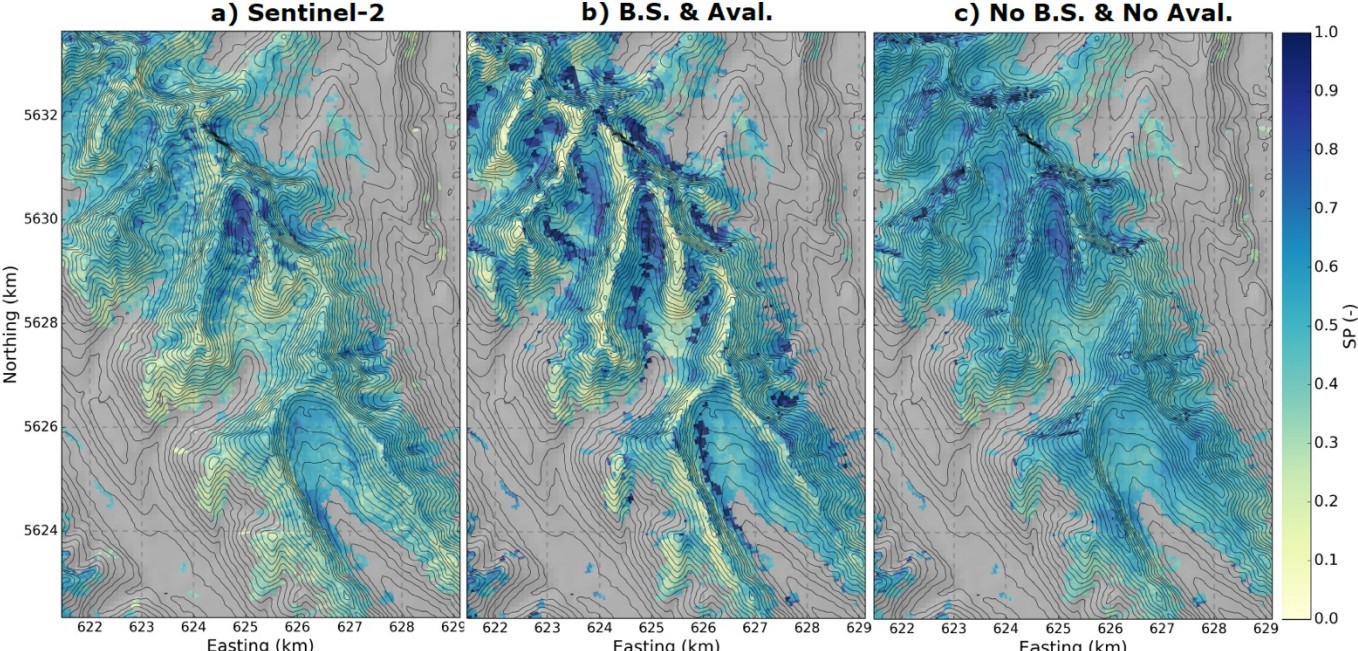

**Figure 10: Maps of snow persistence index (SP) (a) derived from Sentinel-2 and simulated by two CHM configurations: (b) *BS Av Rc*, and (c) *NoBS NoAv* (Table 3). Pixels covered by tall vegetation in the observations and in the simulations are excluded from the comparison and appear in grey. Black isolines correspond to Δz = 50 m and the location of this region in shown on Fig. 1b.**






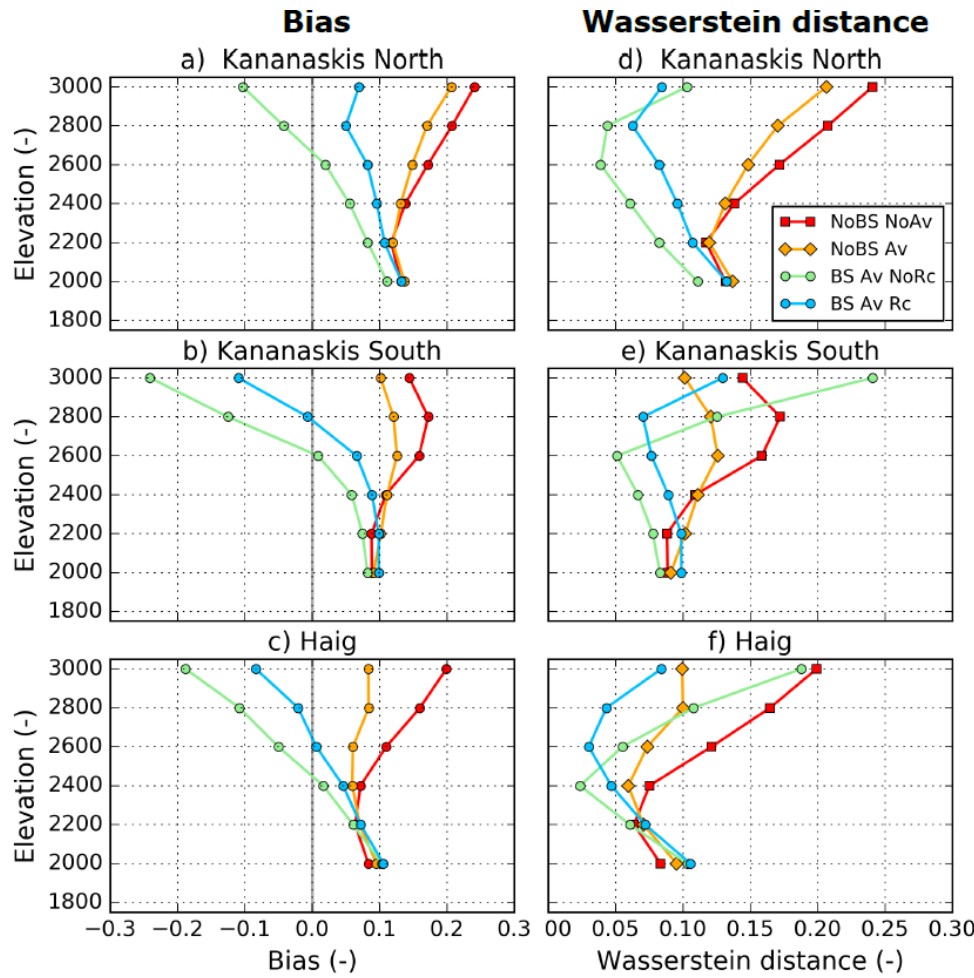

**Figure 11: Bias and Wasserstein distance between observed and simulated snow persistence index as a function of elevation for four CHM experiments and three sub-regions. The location of these sub-regions is shown on Fig. 1.**




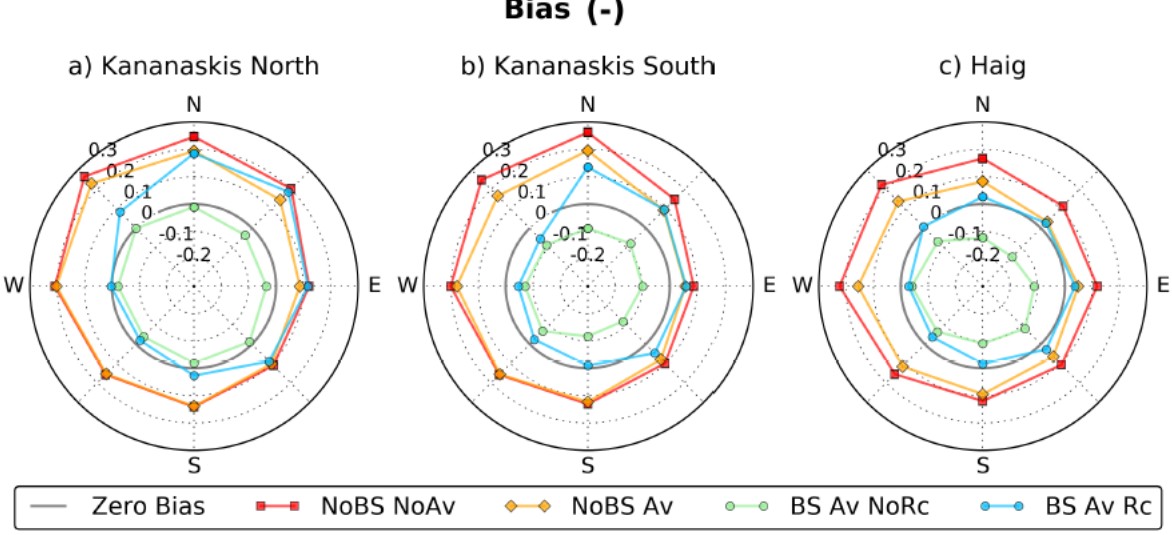

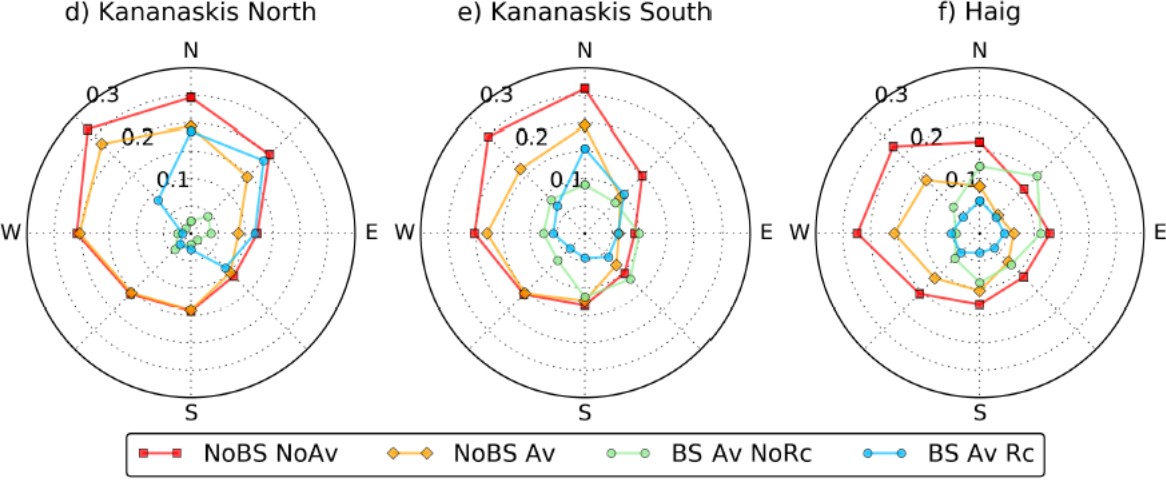

**Figure 11: Same as Fig. 9 for the snow persistence index (SP)**
