# Peer review of "Multi-scale snowdrift-permitting modelling of mountain snowpack"

_The Cryosphere, 2020_

## Referee Comment (RC1) · Rebecca Mott (Referee) · 1 Sep 2020

This study introduces a modelling approach capturing processes which drive snow depth variability at the ridge and mountain range scale. While different models exist capturing snow redistribution processes at very small-scale, this study is able to efficiently model snow redistribution for large domains in a computationally efficient way. The paper is very well written and results are well presented. State-of-the art models and methods are combined to improve the spatial variability of snow depths at the ridge scale. Strengths and limitations of the model approaches are well discussed. Modelled snow redistribution was verified against ALS and SNETINEL-2 data at the end of the winter season. In my opinion a comparison with SNETINEL-2 data covering only the accumulation season would better represent snow distribution affected by snow redistribution processes (without contribution of snow melt). It would also be interesting to see how process representation affects the spatio-temporal snow dynamics. Please find my minor comments below.

Comments: Abstract: The abstract is well-written and concise. The new wind down-scaling strategy is mentioned in the abstract. I would recommend to add one sentence description of this method to abstract to give the reader a rough idea how the method works.

I recommend to better highlight the multi-scale approach by more clearly acknowl-edging the combination of regional-scale weather data and downscaling techniques allowing for snow redistribution modelling.

Only blowing snow is mentioned in the abstract. As you also calculate drifting snow via saltation this process should also be added; Introduction:

P2, 44: I would recommend to add the reference Schlögl et al. 2018 for heat advection processes

P3, L1: I am not convinced that 200 m resolution can be called a snow drift permitting scale, but this is open for discussion;

P 3, p89: you could add here the possibility of modelling preferential deposition in atmospheric models;

P5: please discuss the effect of the number of layers on the availability of erodible snow. Please add more details on whether the model distinguishes between hard and soft snow and which characteristics determine the erodibility of snow – e.g. wetness of snow?

P6, L179: is preferential deposition calculated as part of the suspension layer?

P 6, L 180: here you could also add the effect of snow redistribution by avalanches on glaciers or ice fields (Mott et al., 2019)

P8, L243: why has WindNinja used a bare ground instead of a smoother snow-covered ground? Is there any possibility to initialize WindNinja with a measured snow distribution?

P8, L 178: is 5 m enough to account for suspension plumes which can be much higher than 5 m?? In my opinion this arbitrary model height might be part of the discussion;

P 8: how sensitive is the blowing and drifting snow model to the model resolution? Why was the model resolution set to 50 m and not higher to better capture saltation but also to better resolve the wind field? Due to computational resources?

P 11, L325: please change Grunewald to Grünewald

P 14, L418: strong, spatial —- delete the comma

P 488-489: I do not understand this sentence

P 17, L 520 – again. Is there a reason why the simulations were limited to 50 m? I thought that it is the advantage of the meshed grid to locally allow for very high resolutions. Especially, at the ridges higher resolution could have a large effect

P 17: L 539: mass-conserving

P 17: What is the contribution of snow melt to final snow depth pattern observed by SNETINEL-2 data and ALS in late April. The SENTINEL-2 data (Figure 10) show the snow persistence index SP at the end of winter. There are some slopes with very low SP values where I could imagine that lateral snow redistribution processes are of minor importance for the snow distribution as these might be more affected by melt. I recommend to additionally use mid-winter SP values which would better reflect the contribution of snow redistribution processes. A comparison of SP values at different stages of the winter would be highly interesting to reflect spatio-temporal dynamics.

P 19: please also discuss the uncertainty due to the constant transfer function value fdown of 0.25. I could imagine that this value changes in downwind distance of the

ridge and might be a function of wind speed and atmospheric stability.

Figures 6 and 10: poor visibility of grid lines;

P 21, L 669: In my opinion subgrid topographic effects primarily affect the local flow field which then affect snow redistribution.

P 22, L 678: high-resolution observations of what?

Suggested reference: Schlögl, S., Lehning, M., Fierz, C., & Mott, R. (2018). Representation of horizontal transport processes in snowmelt modeling by applying a footprint approach. Frontiers in Earth Science, 6, 120 (18 pp.). https://doi.org/10.3389/feart.2018.00120

---

## Referee Comment (RC2) · Tobias Sauter (Referee) · 12 Sep 2020

The work of 'Multi-scale snowdrift-permitting modelling of mountain snowpack' by Vionnet el al. deals with the spatial and temporal evolution of snow cover in high mountain areas. The study focuses, as clearly mentioned in the well-structured introduction, the (i) added value of a wind downscaling approach, (ii) the role of lateral snow redistribution, and (iii) the use of remote sensing data. For this purpose, the authors developed a model chain that combines established models and parameterizations. This research design was applied and validated for the Kananaskis Valley in the Canadian Rockies.

The research priority of the study is nicely summarized in the introduction and shows the reader the scientific challenges in this research area. These questions are taken up

throughout the paper and are finally answered in the conclusion. The description of the methods is a little sparse in some parts, but with the given references it can be easily followed and reproduced by an interested reader. Since these are well established methods and approaches, I think that no further work is necessary. Only the wind downscaling approach raised some questions which can be answered with little effort (see comments below).

The model experiments based on a stepwise model falsification are well thought out. However, abbreviations were not catchy for me and led to confusion and I had to scroll back and forth to check with Table 2.

The results of the downscaling and snowpack simulations are well structured and show sufficiently the strengths and weaknesses of the different approaches and experiments. In the subsequent discussion these results are put into context. For me as a reader all questions that came up in the beginning were answered sufficiently. Also nice is chapter 4.4 where the limits of the approach are discussed.

In summary I think the work fits well to 'The Cryosphere'. The structure follows the classical structure and is easy to understand for the reader. Furthermore, I don't see any concerns in the technical realization and the conclusions. These are also supported by good illustrations. Based on this reviwer, I recommend the publication of the study with only minor revisions.

More specific comments

Section: Atmospheric Forcing

P7L212: Precipitation plays a particularly important role in snow dynamics and is difficult to capture in most applications. I don't doubt that the HRDPS sufficiently accounts for the large- scale precipitation effects on average, but don't the strong topographic variations lead to strong subgrid-scale gradients (< 2.5 km), which in turn reduces the variability on the small scale?

Section: Wind field downscaling

The general downscaling approach is comprehensible, and the combination of a wind library and transfer function seems to be reasonable. While reading through the section I asked myself at some points why the following steps were implemented in that way:

(i) Diagnostic wind models are computationally efficient. This efficiency would allow for separate simulations for each time step. Why not following this approach?

(ii) As far as I can see the wind velocity at 40 m above ground was set to 10 m/s for each simulation. Why weren't different wind classes introduced here? In my understanding the background wind has a significant influence on the flow features (e.g. flow separation, gap flow, bluff body formation etc.). Have you checked different boundary conditions?

(iii) As in the study by Barcons et al. (2018) the characteristic length, L, was set to 1000 m. How was this length determined? Do we not expect very different lengths for different topographies? How sensitive are the simulation to this length scale?

(iv) Are the wind fields still mass consistent when two micro-scale wind fields are linearly interpolated? Maybe a mass correction might be necessary.

(v) In the WindNinja model a spatially constant roughness length was assumed, which is due to the nature of the model. Later in the same paragraph it is described that the prognostic wind velocity at 10 m takes into account the interaction with the vegetation by adjusting the logarithmic wind profile. I doubt that surface properties are homogeneous at a horizontal resolution of 30 m. Wouldn't it be useful to consider surface properties of a defined upstream fetch when adjusting the wind speed?

(vi) The fact that mass-consistent models cannot represent flow separation and other flow features is the major deficit of such models. The approach to adapt the transfer function using the Winstral parameter seems to be a good way to start. I just wondered why a value of 0.25 was used for the transfer function. From a fluid dynamic point of
view, flow separation zones usually lead to a flow reversal and not to a reduction of the wind speed. Maybe the simulations could be improved by a dynamic value.

(vii) Due to the limited number and location of stations, there is no real evidence that downscaling leads to an improved characterization of the wind field. However, this could be shown by the means of the snowpack simulations and the comparison with the ALS and Sentinel data. To be more concise, I would recommend a Experiment using the HRDPS simulations directly with the snowdrift scheme and recirculation parametrization (see comment below).

Section: Snowpack simulations

It would be interesting to run the snowpack simulations without wind downscaling but rather drive the snow drift module and recirculation parametrization with the HRDPS fields (without WindNinja). I think it would be helpful for the community to see the importance of high-resolution wind fields.

Minor comments

P3L76: You need commas berfore and after 'inspired by Ryan (1977)'.

P6L174: The abbrevation 'PBSM-3D' has not been introduced.

P14L443: As mentioned in a previous comment it would be useful to correct the HRDPS precipitation.

P16L496: Are these correlations significant?

P17L538: Maybe I missed something, but there is no experiment where the sensitivity of snow drift simulations in CHM is shown without the WindNinja fields.

---

## Author Comment (AC1) · 19 Oct 2020

**Answer to Rebecca Mott TC-2020-187**

We thank Rebecca Mott for her comments. We provide here our responses to those comments and describe how we addressed them in the revised manuscript. The original reviewer comments are in normal black font while our answers appear in blue font.

This study introduces a modelling approach capturing processes which drive snow depth variability at the ridge and mountain range scale. While different models exist capturing snow redistribution processes at very small-scale, this study is able to efficiently model snow redistribution for large domains in a computationally efficient way. The paper is very well written and results are well presented. State-of-the art models and methods are combined to improve the spatial variability of snow depths at the ridge scale. Strengths and limitations of the model approaches are well discussed. Modelled snow redistribution was verified against ALS and SNETINEL-2 data at the end of the winter season. In my opinion a comparison with SNETINEL-2 data covering only the accumulation season would better represent snow distribution affected by snow redistribution processes (without contribution of snow melt). It would also be interesting to see how process representation affects the spatio-temporal snow dynamics. Please find my minor comments below.

Our response to this general comment is detailed below in our answers to the specific comments.

**Comments:**

**Abstract:**
The abstract is well-written and concise. The new wind downscaling strategy is mentioned in the abstract. I would recommend to add one sentence description of this method to abstract to give the reader a rough idea how the method works.
We rewrote the sentence describing the wind downscaling in the abstract to give more details about the method: "*In particular, a new wind downscaling strategy uses pre-computed wind fields from a mass-conserving wind model at 50-m resolution to perturb the meso-scale HRDPS wind and to account for the influence of topographic features on wind direction and speed.*"

I recommend to better highlight the multi-scale approach by more clearly acknowledging the combination of regional-scale weather data and downscaling techniques allowing for snow redistribution modelling.
The multi-scale approach is now highlighted at the beginning of the abstract after the description of the main processes causing the spatial variability in snow accumulation and ablation:
"*The multi-scale approach combines atmospheric data from a numerical weather prediction system at km-scale with process-based downscaling techniques to drive the Canadian Hydrological Model (CHM) at spatial resolutions allowing for explicit snow redistribution modelling.*"

Only blowing snow is mentioned in the abstract. As you also calculate drifting snow via saltation this process should also be added;
We prefer the term blowing snow for the whole saltation and suspension transport, rather than drifting snow which is not well defined. This is now clarified in the abstract and the introduction.

Introduction:
P2, 44: I would recommend to add the reference Schlögl et al. 2018 for heat advection Processes
A good suggestion. Reference now added to the text in the revised version of the manuscript.

P3, L1: I am not convinced that 200 m resolution can be called a snow drift permitting scale, but this is open for discussion;
We mention the resolution of 200 m in the introduction here since this resolution has been used in two studies by Bernhardt et al. (2009 and 2010) focusing on high-resolution modelling of wind-induced snow transport in alpine terrain. Bernhardt et al (2010) compared two resolutions (200 m and 30 m) and showed that the simulated snow redistribution was quite sensitive to the choice of the resolution with an

overestimation of snow transport at 200 m. This lower resolution corresponds to the upper range of snowdrift permitting scales (defined as the range of resolutions that requires the activation of horizontal snow redistribution between computational element). A proper treatment of blowing snow at this resolution requires accounting for subgrid processes which are missing in the main models simulating blowing snow (as discussed in Sect 4.4). The resolution of 200 m used in Bernhardt et al (2010) is now explicitly mentioned at the end of page 3 when describing snowdrift-permitting models that can run over entire snow seasons.

P 3, p89: you could add here the possibility of modelling preferential deposition in atmospheric models;
Preferential deposition is now mentioned in the introduction as follows:
*"These advanced models can be used for detailed studies such as the feedbacks between blowing snow sublimation and the atmosphere (Groot Zwaaftink et al., 2011) or the processes driving the variability of snow accumulation during a snowfall event, including preferential deposition of snowfall (Lehning et al., 2008; Mott et al., 2010; Vionnet et al., 2017)."*

P5: please discuss the effect of the number of layers on the availability of erodible snow. Please add more details on whether the model distinguishes between hard and soft snow and which characteristics determine the erodibility of snow – e.g. wetness of snow?
PBSM-3D uses the formulation of Li and Pomeroy (1997) to compute the threshold wind speed for blowing snow. This formulation is based on a large, multiyear field dataset and depend on air temperature and snowpack presence. This approach is also used in the CRHM model (Pomeroy et al., 2007) and performs well for blowing snow simulation studies in the Canadian Rockies (e.g. MacDonald et al, 2010) and elsewhere, but is not coupled to the snow properties simulated by Snobal such as density and liquid water content.
The text of the revised manuscript has been modified has followed:
- In Sect. 2.2 describing CHM: "*It deploys the parameterization of Li and Pomeroy (1997) to determine the threshold wind speed as a function of air temperature and **snow presence. It is not coupled to the properties of surface snow (e.g. density, liquid water content) simulated by Snobal (see Sect 4.4 for a discussion on the limitation of this approach).***"
- In Sect. 4.4 discussing the limitations of the model: "*In addition, CHM uses the formulations for the threshold friction velocity of Li and Pomeroy (1997) that only depend on snow presence, and air temperature**. Though based on a large multi-year observational dataset, such parameterization is empirical and** does not also account for the effect of snow fragmentation during blowing snow events (Comola et al., 2017) which may lead to an underestimation of the threshold friction velocity and an overestimation of blowing snow occurrence in alpine terrain (Vionnet et al., 2013). **CHM may benefit from the inclusion of a more physically based snow transport routine in the future.***"

P6, L179: is preferential deposition calculated as part of the suspension layer?
Preferential deposition of snowfall is not directly represented in CHM, apart from how it is addressed in the GEM atmospheric model microphysics scheme as part of snowfall calculations. In CHM, new snowfall is directly added at the top of the snowpack in Snobal and does not initially interact with blowing snow in the suspension layer. Instead, if the wind speed is larger than the threshold wind speed for snow transport, new snowfall is redistributed through both saltation and suspension blowing snow layers. The absence of explicit simulation of preferential deposition in CHM was already discussed in Sect 4.4. We added a sentence in Sect. 2.2.2 in the revised manuscript:
*"Snowfall over complex terrain is calculated by GEM according to its microphysics scheme (Milbrandt et al., 2016). CHM does not simulate explicitly preferential deposition of snowfall (Lehning et al., 2008; Mott et al., 2018). New snow is added to the surface layer in Snobal and, if wind speeds exceed the threshold wind speed, it is transported in the saltation and suspension blowing snow layers by PBSM-3D"*

P 6, L 180: here you could also add the effect of snow redistribution by avalanches on glaciers or ice fields (Mott et al., 2019)
This reference has been added to the text as follows:

*"In steep alpine terrain, gravitational snow transport strongly affects the spatial variability of the snowpack (e.g. Sommer et al., 2015),* **the mass balance of glaciers** *(Mott et al., 2019) and modifies the runoff behaviour of alpine basins (Warscher et al. 2013)."*

P8, L243: why has WindNinja used a bare ground instead of a smoother snow-covered ground? Is there any possibility to initialize WindNinja with a measured snow distribution?
In the wind simulations used in this paper, WindNinja has been initiated with a homogenous surface with a constant aerodynamic roughness of 0.01m. This value is the same as the one used to represent a snow-covered topography in alpine terrain with the ARPS atmospheric model in Mott et al. (2010) and Mott and Lehning (2010). In its present software version, WindNinja cannot be initialized with a measured snow distribution that would locally perturb the snow/atmosphere energy and momentum exchanges.
The following sentence has been modified in Sect. 2.2.4:
*"WindNinja used a spatially constant roughness length ($z_0 = 0.01$ m)* **representative of snow-covered terrain in alpine topography** *(Mott et al., 2010; Mott and Lehning, 2010) and vegetation effects ..."*

P8, L 178: is 5 m enough to account for suspension plumes which can be much higher than 5 m?? In my opinion this arbitrary model height might be part of the discussion;
We selected 5 m for PBSM-3D since this height is usually sufficient to capture most of the mass transported in the suspension layer (see Figure 1 below) due to the strong vertical gradient of blowing snow concentration close to the surface. In steady state, 10-m wind speeds above 15 m s$^{-1}$ are required to obtain a contribution larger than 10% for the part of the suspension layer above 5 m (Fig. 1) This contribution raises quickly above 15 m s$^{-1}$. However, the steady-state approximation implies fetches greater than 300 m that are rarely encountered in alpine terrain so that transport rates in alpine terrain are usually lower than the steady state approximation for the same wind speed (Naaim Bouvet et al., 2010). Therefore, we estimate that a height of 5-m is sufficient to capture most of the mass of snow transported by the wind over slopes exposed to the wind in alpine terrain.
On the other hand, we agree with the reviewer that such configuration may not fully capture suspension plumes, especially those forming above the leeward side of crest lines due to a massive advection of snow particles in the free atmosphere. In CHM, all the snow in the suspension layer that is transported across a crest remains a maximum height of 5 m above the ground and will eventually be deposited due to lower wind speed on the leeside of the crest. Such behavior is unrealistic and could potentially be improved by assuming a thicker suspension layer in CHM. A proper modelling of suspension plumes would ultimately require a three-dimensional wind field and calculation of flow separation in the lee of the crest.
In the revised manuscript we point out the limitation associated with the 5-m thickness for the modeling of snow plumes above alpine ridges:
*"Finally, CHM uses a thickness of 5-m for the suspension layer (Marsh et al., 2020a). This is sufficient to capture most of the mass transported in alpine terrain over slopes exposed to wind with limited fetches (MacDonald et al., 2010; Naaim Bouvet et al, 2010) but it cannot simulate the formation of snow plumes at crest lines. Mass loss due to the advection of blown snow particles to atmospheric layers and subsequent sublimation are likely underestimated by CHM."*

[Figure]

*Figure 1: Relative contribution of the saltation and suspension layers to the total transported mass in steady state. For the suspension layer, the contributions below and above 5 m are considered separately. The mass concentration in the saltation and suspension layers were obtained from Pomeroy and Male (1992) assuming a roughness length of 0.01 m, a threshold wind speed of 0.25 m $s^{-1}$ and a constant settling velocity of 0.5 m $s^{-1}$.*

P 8: how sensitive is the blowing and drifting snow model to the model resolution? Why was the model resolution set to 50 m and not higher to better capture saltation but also to better resolve the wind field? Due to computational resources?

These are pertinent questions raised by the reviewer. We did not quantify the impact of mesh resolution on simulated snow redistribution in our study. In the paper describing PBSM-3D, however, Marsh et al (2020) show that a coarse discretization averaged out variability and removed the small-scale variability in areas of the simulation domain covered by large triangles (area of 300 $m^2$ for the coarse triangles VS 3 $m^2$ for the high- and fixed resolution reference mesh). We expect a similar sensitivity in our simulation domain and a higher mesh resolution would certainly reduce the overestimation of snow transport from windward to leeward slopes as reported in Mott and Lehning (2010) using a different snow redistribution model. A minimal triangle size of 50-m was used in our study to reach snowdrift-permitting scales and yet achieve reasonable computational time with CHM (less than 5h on 32 processors for the whole simulation). This 50-m resolution could be adopted for future operational basin-scale snowpack simulations in the Canadian Rockies.

The resolution of Wind Ninja has been set to 50 m to match the minimal triangle size and was not imposed by computational resources since the wind library is pre-computed before the CHM simulations. In our case, the maximal resolution of WindNinja could have been 30 m, corresponding to the original resolution of SRTM DEM used in this study.

P 11, L325: please change Grunewald to Grünewald
Changed

P 14, L418: strong, spatial —- delete the comma
Corrected

P 488-489: I do not understand this sentence
This sentence has been rephrased as follows:
*"These results are consistent with the overestimation of gravitational snow redistribution to lower elevation and the erroneous location of avalanche deposits observed on Fig. 6b."*

P 17, L 520 – again. Is there a reason why the simulations were limited to 50 m? I thought that it is the advantage of the meshed grid to locally allow for very high resolutions. Especially, at the ridges higher resolution could have a large effect

The choice of the 50-m resolution was explained in our answer to a previous question.

P 17: L 539: mass-conserving

Corrected.

P 17: What is the contribution of snow melt to final snow depth pattern observed by SNETINEL-2 data and ALS in late April. The SENTINEL-2 data (Figure 10) show the snow persistence index SP at the end of winter. There are some slopes with very low SP values where I could imagine that lateral snow redistribution processes are of minor importance for the snow distribution as these might be more affected by melt. I recommend to additionally use mid-winter SP values which would better reflect the contribution of snow redistribution processes. A comparison of SP values at different stages of the winter would be highly interesting to reflect spatio-temporal dynamics.

The snow depth measured was measured by ALS on 27$^{th}$ April 2018. This date corresponds approximately to the peak snow accumulation over the domain (see Fig. 2 below).

We agree that the Sentinel-2 SP index reflects the spatial heterogeneities of both ablation and accumulation processes. In our paper, however, we compared different model experiments which only differed in their representation of the snow transport processes, while the melt algorithm remained unchanged. Differences in the simulated SP thus only result from the snow transport parameterization. The Sentinel-2 SP index was computed using images from 1$^{st}$ April 2018 to 31$^{st}$ August 2018. SP is suitable to identify areas that remain preferentially snow covered during the spring and the summer such as drifts or avalanche deposits due to large snow accumulation and areas with low ablations such Northern shaded areas. In wintertime, most of the pixels are covered by snow, even on windward slope exposed to wind-induced snow transport, so that a mid-winter SP (computed for example from 1$^{st}$ November) gives values close to 1 over most of the domain. To avoid this limitation, Wayand et al. (2018) proposed a snow absence (SA) index calculated from snow-free areas during the winter to identify areas of wind-erosion or avalanche source areas. However, the identification of such regions with SA is challenging since a few cm of snow are sufficient to cover them. For these reasons, mid-winter SP and SA were not considered in this study since they do not bring enough valuable information to evaluate CHM simulations. Overall, SP can only provide an integrative view of the spatiotemporal dynamics since it is influenced by variability in both snow ablation and accumulation (Wayand et al. 2018). At the moment, only successive ALS measurements can provide information on spatio-temporal snow dynamics at the basin scale (e.g. Hedrick et al., 2018).

A sentence has been added in Sect 4.2 to highlight the importance of accounting for variable insolation effects when using the snow persistence index to evaluate snow redistribution models:
*"As illustrated by Wayand et al. (2018), the snow persistence index is influenced by variability in both snow accumulation and ablation, so that this index can only be used to evaluate snow redistribution*

*models if variable insolation effects are also simulated. This is the case in the simulations presented in this paper (Sect 2.2.5)."*

[Figure]

*Figure 2: Temporal evolution of SWE in open terrain per elevation band for two CHM experiment. The vertical orange line shows the date when the ALS winter scan was collected (27 April 2018).*

P 19: please also discuss the uncertainty due to the constant transfer function value fdown of 0.25. I could imagine that this value changes in downwind distance of the ridge and might be a function of wind speed and atmospheric stability.

The constant value of 0.25 used in this study in based on the initial developments of Winstral et al. (2009). The recent study of Menke et al. (2019) has measured the ratio, $R$, between the maximum wind speed in recirculation zones and the wind speed in the inflow at the crest. This ratio is similar to $f_{down}$ used in our downscaling method to reduce the wind speed in areas prone to flow recirculation. Figure 10 in Menke et al. (2019) shows how $R$ depends on the Richardson number (used to quantify the atmospheric stability). Their results show that $R$ typically ranges between 0.1 and 0.5 for unstable atmospheric conditions and between 0.05 and 0.35 for stable atmospheric conditions. $R$ tends to decrease with increasing stability in a stable atmosphere. As reported by Menke et al (2019), a ratio of less than 0.3 are observed for wind speed greater than 12 m s$^{-1}$, characterized by neutral or slightly stable atmospheric conditions. The study of Menke et al (2019) is mentioned in Sect. 4.3 in the revised version of the paper:

*"A constant value of 0.25 is used for the transfer function in recirculation zones. This value falls within the range of values reported on Fig. 10 of Menke et al. (2019) for the ratio, R, between the maximum wind speed in recirculation flow and the inflow wind speed at the crest. Menke et al. (2019) found that R tends to decrease with increasing stability in a stable atmosphere and it presents values lower than 0.3 for inflow wind speed greater than 12 m s$^{-1}$. This suggests that a dynamic value based on atmospheric stability could be used for the transfer function in recirculation zones."*

Figures 6 and 10: poor visibility of grid lines;
The figures were modified to better show the grid lines.

P 21, L 669: In my opinion subgrid topographic effects primarily affect the local flow field which then affect snow redistribution.

We fully agree with the reviewer and modified the conclusion accordingly:

*"This is potentially due to the absence of subgrid topographic effects **in the driving wind field and in the snow transport equations** in CHM."*

P 22, L 678: high-resolution observations of what?

Thanks for noticing it. In the revised version of the manuscript we use: *"high-resolution observations such as ALS snow depth or Sentinel-2 snow cover."*

Suggested reference:

Schlögl, S., Lehning, M., Fierz, C., & Mott, R. (2018). Representation of horizontal transport processes in snowmelt modeling by applying a footprint approach. Frontiers in Earth Science, 6, 120 (18 pp.). https://doi.org/10.3389/feart.2018.00120

References

Menke, R., Vasiljević, N., Mann, J., and Lundquist, J. K.: Characterization of flow recirculation zones at the Perdigão site using multi-lidar measurements, Atmos. Chem. Phys., 19, 2713–2723, https://doi.org/10.5194/acp-19-2713-2019, 2019.

---

## Author Comment (AC2) · 19 Oct 2020

Answer to Tobias Sauter TC-2020-187

We thank Tobias Sauter for his comments. We provide our responses to his comments and describe how we addressed them in the revised manuscript. The original reviewer comments are in normal black font while our answers appear in blue font.

The work of 'Multi-scale snowdrift-permitting modelling of mountain snowpack' by Vionnet el al. deals with the spatial and temporal evolution of snow cover in high mountain areas. The study focuses, as clearly mentioned in the well-structured introduction, the (i) added value of a wind downscaling approach, (ii) the role of lateral snow redistribution, and (iii) the use of remote sensing data. For this purpose, the authors developed a model chain that combines established models and parameterizations. This research design was applied and validated for the Kananaskis Valley in the Canadian Rockies.

The research priority of the study is nicely summerized in the introduction and shows the reader the scientific challenges in this research area. These questions are taken up throughout the paper and are finally answered in the conclusion. The description of the methods is a little sparse in some parts, but with the given references it can be easily followed and reproduced by an interested reader. Since these are well established methods and approaches, I think that no further work is necessary. Only the wind downscaling approach raised some questions which can be answered with little effort (see comments below).

The model experiments based on a stepwise model falsification are well thought out. However, abbreviations were not catchy for me and led to confusion and I had to scroll back and forth to check with Table 2.

The results of the downscaling and snowpack simulations are well structured and show sufficiently the strengths and weaknesses of the different approaches and experiments. In the subsequent discussion these results are put into context. For me as a reader all questions that came up in the beginning were answered sufficiently. Also nice is chapter 4.4 where the limits of the approach are discussed.

In summary I think the work fits well to 'The Cryosphere'. The structure follows the classical structure and is easy to understand for the reader. Furthermore, I don't see any concerns in the technical realization and the conclusions. These are also supported by good illustrations. Based on this reviwer, I recommend the publication of the study with only minor revisions.

The names of the model experiments have been modified in Table 2, in the text and in the figures. We hope it will reduce the confusion mentioned by the reviwer.

More specific comments

Section: Atmospheric Forcing

P7L212: Precipitation plays a particularly important role in snow dynamics and is difficult to capture in most applications. I don't doubt that the HRDPS sufficiently accounts for the large- scale precipitation effects on average, but don't the strong topographic variations lead to strong subgrid-scale gradients (< 2.5 km), which in turn reduces the variability on the small scale?

As mentioned by the reviewer, the configuration of CHM used in this paper does not account for the spatial variability in snowfall amount at spatial scales below 2.5 km (the HRDPS resolution) which impacts the simulated small-scale variability of snow depth. At these scales, the spatial variability in snowfall amount results from: (i) snowfall enhancement caused by the interaction of the flow field with the local topography and local cloud formation processes, such as seeder-feeder mechanisms and; (ii) pure particle flow interaction (preferential deposition of snowfall) (e.g. Mott et al., 2018). So far, these local processes have been previously studied using computationally expensive 3-D atmospheric models at high-resolution (below 50-m) that can explicitly simulate these processes (e.g., Mott et al., 2010;

Dadic et al., 2010; Vionnet et al., 2017; Gerber et al., 2019). In the context of 2D distributed snowpack modelling, such processes cannot be directly simulated. Two main approaches can be tested: (i) a precipitation adjustment function depending on the differences between the elevation of the 2.5 km model and the elevation of the high-resolution CHM mesh (Thornton et al., 1997; Liston and Elder, 2006) and; (ii) a parameterization for preferential deposition of snowfall (Dadic et al., 2010).

The precipitation adjustment function was tested by Vionnet et al. (2019) where they downscaled NWP forecast from 2.5 km to 500-m grid spacing in the French Alps. They showed poor performances at high elevations that can be partially related to the value of the precipitation-elevation adjustment factor used in Liston and Elder (2006). The same method was tested to downscale the HRDPS precipitation amount to the high-resolution CHM mesh over the Kananaskis domain. The impact on the simulated snow depth as a function of elevation for three sub-regions is shown on Fig. 1 below. The CHM simulations shown on this figure do not include blowing snow and gravitational redistribution. Introducing the adjustment factor leads to a continuous increase in snow depth as a function of elevation. In particular, compared to the default HRDPS precipitation, larger snow depths are found above 2300 m which correspond to areas that are higher than the HRDPS grid. This shows that accounting for sub-grid effects on snowfall amount can strongly impact the elevation-dependency of snow depth.

The precipitation adjustment function was initially developed to generate a distributed precipitation field accounting for topographic effect from sparse measurements in mountainous terrain. However, the degraded results shown in Fig 1 from including it suggest that it may not be suitable for capturing sub-grid effects within a 2.5-km grid. Therefore, we did not include this correction due to the additional uncertainty that it would introduce.

The parameterization of preferential deposition of snowfall proposed by Dadic et al. (2010) requires estimations of the horizontal velocity as well as the vertical velocity (Eq. 2 in Dadic et al. (2010)). The estimation of the vertical velocity is not included in CHM, and so that this parameterization cannot be implemented. Obtaining the vertical velocity from WindNinja simulations could be useful to drive this parameterization in future studies.

The following sentences were added in the revised manuscript:
- Section 2.2.3: "T*he precipitation adjustment function of Liston and Elder (2006) has been tested but it led to strong overestimation of snow depth at high elevation (not shown), suggesting that this factor may not be adapted to account for the subgrid variability of precipitation amount within a 2.5 km grid.*"
- *Section 2.2.2: "CHM does not simulate explicitly preferential deposition of snowfall (Lehning et al., 2008; Mott et al., 2018)."*
- Section 4.4: *"The parameterization of Dadic et al. (2010) could be tested in CHM but would require an estimation of the vertical wind speed that could be provided by WindNinja"*

[Figure]

*Figure 1: Boxplots showing the distribution of observed and simulated snow depth per 100-m elevation bands for three sub-regions and two CHM simulations. HRDPS default: no correction of the precipitation amount; HRDPS Thornton: Adjustment of the precipitation amount using the correction factor proposed in Thornton et al. (1997).*

Section: Wind field downscaling

The general downscaling approach is comprehensible, and the combination of a wind library and transfer function seems to be reasonable. While reading through the section I asked myself at some points why the following steps were implemented in that way:

(i) Diagnostic wind models are computationally efficient. This efficiency would allow for separate simulations for each time step. Why not following this approach?
Wind Ninja is a computationally efficient wind model compared to a more complex model such as a CFD-model or an atmospheric model in Large Eddy Simulation mode. Nonetheless, in the context of this work, running CHM over a full snow season at an hourly time step requires 8760 (24*365) distributed driving wind fields. In a different study (in preparation), WindNinja was used to downscale separately the 8760 low-resolution HRDPS wind fields to 50-m resolution over the 1000 km² of the Kananaskis domain following the method described in Wagenbrenner et al. (2016). It took almost 19 days of wall-clock time whereas the downscaling method proposed in our paper took 4.5 hours of wall-clock time. Therefore, our approach brings a substantial improvement (100x) in computational cost. An article comparing the two downscaling methods is in preparation.

(ii) As far as I can see the wind velocity at 40 m above ground was set to 10 m/s for each simulation. Why weren't different wind classes introduced here? In my understanding the background wind has a significant influence on the flow features (e.g. flow separation, gap flow, bluff body formation etc.). Have you checked different boundary conditions?

In the initial development of the downscaling method, WindNinja simulations were carried out over the Kananaskis domain using different input wind speeds. The idea was to generate a wind field library containing different large-scale wind speeds as in Mott et al. (2010). However, results showed that very similar transfer functions (Eq 1 in the paper) were derived from these different WindNinja experiments. We suspect that it results from a linear behavior of the WindNinja solver. For this reason, only one input wind speed was used when building the wind library as in Barcons et al. (2018).

Using different wind classes would be highly relevant if a mass- and momentum-conserving model was used to build the wind library. Indeed, such a model would be able to simulate significant flow features that depend on the intensity of the background wind, in particular the formation of recirculation zones and their spatial extension.

Information we have added in the revised manuscript on this subject:

- Section 2.2.4: "*Only one value for the initial wind speed was used to build the wind library due to the insensitivity of the transfer function to the initial wind speed found with WindNinja.*"
- Section 4.3: "*Improvements in the wind downscaling could be achieved using such models to generate the library of wind fields, as proposed by Barcons et al. (2018). Different conditions of atmospheric stability could also be considered (e.g., Gerber et al., 2017) as well as different input wind speeds that affect significant flow features such as flow separation.*"

(iii) As in the study by Barcons et al. (2018) the characteristic length, L, was set to 1000 m. How was this length determined? Do we not expect very different lengths for different topographies? How sensitive are the simulation to this length scale?

Barcons et al. (2018) determined that a circle of radius of 500 m (averaging area of 0.78 km$^2$) gave the optimal performance (RMSE, Skill) for their downscaled wind field when compared to wind mast observations. They downscaled WRF output at 3-km grid spacing over complex terrain in Mexico. This resolution is similar to the 2.5 km resolution of the HRDPS used in our paper. For this reason, we decided to use a similar averaging area than Barcons et al. (2018). For computational reasons, we decided to adopt a square instead of a circle for the shape of the averaging area and ultimately selected an area of 1 km$^2$ (characteristic length, L = 1 km).

We fully agree with the reviewer that the optimal value of this characteristic length certainly depends on the complexity of the topography as well as the initial resolution of the input wind field. Indeed, the maximal value for the characteristic length is the resolution of the input wind field since above this value the transfer function starts including features that are already resolved in the input wind field. Conceptually, this characteristic length should be large enough to cover the distance between the main sub-grid topographic features that are not captured in the input wind field. High-resolution wind simulations (for example at 50-m resolution) could be used to study the dependency of the characteristic length on the complexity of the terrain.

The choice of characteristic length, $L$, influences the spatial variability of the wind speed. As $L$ increases, the transfer function incorporates the local wind fluctuation induced by the micro-scale terrain features. Figure 2 shows an example of near-surface wind field obtained when downscaling the HRDPS wind field with three values of $L$ (0.5, 1 and 2.5 km). By construction of the downscaling approach, the wind direction is not influenced by the value of $L$. On the contrary, the differences between the downscaled and the HRDPS mesoscale wind speed (shown on Fig. 3 in the manuscript) depend on the value of $L$. The smaller the value of $L$, the more the downscaled wind speed coincides with the HRDPS wind speed. In contrary, as $L$ increases, the transfer function includes more local wind fluctuations around the HRDPS mesoscale wind field. This can be observed around ridges where the downscaled wind speed is larger with $L$ = 2500 m than with $L$ = 500 m and 1000 m.

[Figure]

*Figure 2: Near-surface wind field on 10 September 2017 at 18 UTC from HRDPS downscaled to the CHM mesh using WindNinja (HRDPS+WN). The parameterization for the formation of recirculation zones on leeward slopes is not used here. Three values for the averaging length, L, are tested to compute the transfer function: (a) 500 m, (b) 1000 m and (c) 2500 m.*

Based on Fig. 2, we can expect a strong impact of the characteristic length on the simulated snow redistribution in the upper slopes. This was not investigated in the context of this paper since we focused our study on the impact of process representation on snowpack simulations at snowdrift-permitting scales and the development of a relevant evaluation framework. A future study on the impact of characteristic length on snow redistribution at the mountain range scale would be highly relevant. The following sentence notes this sensitivity in the revised manuscript (Section 4.3):

*"Sensitivity tests revealed that the wind field in the upper slopes strongly depend on the value of the radius of influence with a potential large impact on simulated snow redistribution."*

(iv) Are the wind fields still mass consistent when two micro-scale wind fields are linearly interpolated? Maybe a mass correction might be necessary.

This downscaling approach does not include a mass term. There is no inward or outward flux of air from a cell. This is true for the method detailed here, as well as other 'speed-up' style approaches such as Liston and Elder (2006), Essery (1999) and Barcons et al., (2018). Certainly, the inverse problem could be constructed so-as to determine what initial condition would be required from WindNinja to produce the final output. However, this would almost certainly be numerically ill-posed. The approach detailed here is designed to be an approximation to the underlying mass conserving CFD simulation, and so if such an inverse problem were constructed the velocity field would almost certainly show some discrepancy in mass, i.e., it would not be divergence-free. Specifically, the mass conservation can be affected at two stages due to interpolation:

- Prior to the simulation, when applying the rasters of the wind library (u and v winds components, transfer function) to the triangles of the unstructured mesh using the *mesher* code (Marsh et al., 2018).
- At CHM runtime, when linearly interpolating and recombining the selected microscale wind components including the local terrain effect to obtain the downscaled wind direction

If a mass correction was applied, it should be done at runtime on the unstructured mesh used by CHM. This could substantially increase the computational time of the simulation and remove part of the benefit of the downscaling approach. That is, it would almost certainly be more accurate and more correct to track the flux of mass across computational cells, which would imply running the full CFD model.

(v) In the WindNinja model a spatially constant roughness length was assumed, which is due to the nature of the model. Later in the same paragraph it is described that the prognostic wind velocity at 10 m takes into account the interaction with the vegetation by adjusting the logarithmic wind profile. I doubt that surface properties are homogeneous at a horizontal resolution of 30 m. Wouldn't it be useful to consider surface properties of a defined upstream fetch when adjusting the wind speed?

We agree that they are limitations in the representation of the interactions between the near-surface wind field and the vegetation in the downscaling approach proposed in our study. So far, fetch effects due to the presence of upstream vegetation are not taken into account when adjusting the wind speed. But they are included in the blowing snow redistribution scheme as described in Marsh et al. (2020). The mass concentration in the saltation layer is reduced in regions where flow is developing. PBSM-3D would benefit from a more accurate representation of the wind speed in these regions. The best solution is certainly to account for a spatially variable vegetation cover (and associated roughness) directly in Wind Ninja.

Information have added in the revised manuscript on this subject:

- Section 2.2.2: *"Upwind fetch is calculated for each triangle of the mesh using the fetchr parameterization of Lapen and Martz (1993)* **and is used to reduce the mass concentration in the saltation layer in regions where flow is developing***."*
- *Section 2.2.4: "Wind speeds were then adjusted to 10-m wind speeds using the Prandtl-von Kármán log-linear wind profile and modified to include vegetation interactions using the vegetation cover of the triangle as defined in Sect 2.2.1.* **Fetch effects due to the presence of upstream vegetation are not taken into account when adjusting the wind speed***."*

(vi) The fact that mass-consistent models cannot represent flow separation and other flow features is the major deficit of such models. The approach to adapt the transfer function using the Winstral parameter seems to be a good way to start. I just wondered why a value of 0.25 was used for the transfer function. From a fluid dynamic point of view, flow separation zones usually lead to a flow reversal and not to a reduction of the wind speed. Maybe the simulations could be improved by a dynamic value.

The constant value of 0.25 used in this study in based on the initial developments of Winstral et al. (2009). It was taken as the average value from Eq. 6 in Winstral et al (2009) for values of Sx between 21.5 and 30° that characterize the reduction of wind speed found for this range of values of Sx.

We agree with the reviewer that flow separation zones usually lead to a flow reversal (e.g. Raderschall et al., 2008; Gerber et al., 2017). However, the wind speed in reversal zone is usually lower than the wind speed at the crest. The transfer function using the Winstral parameter aims at capturing this effect. The recent study of Menke et al. (2019) has measured the ratio, $R$, between the maximum wind speed in recirculation zones and the wind speed in the inflow at a crest. This ratio is similar to $f_{down}$ used in our downscaling method to reduce the wind speed in areas prone to flow recirculation. Figure 10 in Menke et al. (2019) shows how $R$ depends on the Richardson number (used to quantify the atmospheric stability). Their results show that $R$ typically ranges between 0.1 and 0.5 for unstable atmospheric conditions and between 0.05 and 0.35 for stable atmospheric conditions. $R$ tends to decrease with increasing stability in a stable atmosphere. As reported by Menke et al (2019), ratio of less than 0.3 are observed for wind speed greater than 12 m s$^{-1}$, characterized by neutral or slightly stable atmospheric conditions. The study of Menke et al (2019) is mentioned in Sect. 4.3 in the revised version of the paper:

*"A constant value of 0.25 is used for the transfer function in recirculation zones. This value falls within the range of values reported on Fig. 10 of Menke et al. (2019) for the ratio, R, between the maximum wind speed in recirculation flow and the inflow wind speed at the crest. Menke et al. (2019) found that R tends to decrease with increasing stability in a stable atmosphere and it presents values lower than 0.3 for inflow wind speed greater than 12 m s$^{-1}$. This suggests that a dynamic value based on atmospheric stability could be used for the transfer function in recirculation zones."*

The absence of modification of the wind direction in the recirculation zones was already discussed in the initial version of the manuscript and it was kept unchanged in the revised version of the paper.

(vii) Due to the limited number and location of stations, there is no real evidence that downscaling leads to an improved characterization of the wind field. However, this could be shown by the means of the snowpack simulations and the comparison with the ALS and Sentinel data. To be more concise, I would recommend a Experiment using the HRDPS simulations directly with the snowdrift scheme and recirculation parametrization (see comment below).

We agree that the evaluation of simulations of snow redistribution driven by different wind fields using distributed snow observations provides only indirect information on the quality of the driving wind field. The reverse is also true and has been explored for alpine ridges by Musselman et al. (2015) and so the relationship between the quality of the wind field and the quality of the snow redistribution field is not straightforward. We used this method in our paper to assess the role and relevance of the parameterization of the wind speed reduction in leeward areas. However, we do not believe that a CHM simulation driven by HRDPS wind fields simply interpolated to the CHM mesh and including the recirculation parameterization will bring new results for the community. Indeed, such experiment would result in a spatially homogenous wind field (except on leeward area) that does not include the wind perturbations generated by the topography at the resolution of the CHM simulations. All the studies using snowdrift permitting models (mentioned in the introduction of our paper) included a minimal downscaling step to account for the effect of the local topography on the wind field at the resolution of the simulation. Indeed, the simulated snow transport and redistribution is a direct consequence of the spatial variability of the wind field (e.g., Musselman et al., 2015). Based on the literature, the minimal topographic adjustment than can be applied to the input wind field is the correction using terrain-based parameters implemented in Liston et al. (2007).

In our paper, the downscaling with WindNinja can be considered as a necessary step prior to any simulation of wind-induced snow redistribution with CHM. It would be relevant to compare the simulated snow redistribution obtained with the HRDPS+WN+Rc wind fields with the redistribution obtained with the Liston et al (2007) approach. This will be the topic of a future study.

Section: Snowpack simulations

It would be interesting to run the snowpack simulations without wind downscaling but rather drive the snow drift module and recirculation parametrization with the HRDPS fields (without WindNinja). I think it would be helpful for the community to see the importance of high-resolution wind fields.

As discussed in our answer to the previous comment, the importance of high-resolution wind fields to drive snow-redistribution models has been already shown in many previous studies mentioned in the introduction (e.g. Gauer, 1998; Liston et al., 2007; Lehning et al., 2008; Bernhardt et al., 2010; Mott and Lehning, 2010; Schneiderbauer and Prokop, 2011; Sauter et al., 2013; Musselman et al., 2015; Vionnet et al., 2014; 2017). For this reason, we opted to not include the additional simulation recommended by the reviewer. Instead, we think that the next step to our study would be to extend the studies of Mott and Lehning (2010) and Musselman et al (2015) and to evaluate in detail the impact of different downscaling method and resolution on snow redistribution at the mountain range scale.

Minor comments
P3L76: You need commas berfore and after 'inspired by Ryan (1977)'.
Corrected.

P6L174: The abbrevation 'PBSM-3D' has not been introduced.
The abbreviation is now defined at the beginning of the paragraph describing the drifting and blowing snow scheme implemented in CHM:
*"CHM also includes a 3-D advection-diffusion blowing snow transport and sublimation model (Marsh et al., 2020a): the 3-D Prairie Blowing Snow Model (PBSM-3D)."*

P14L443: As mentioned in a previous comment it would be useful to correct the HRDPS precipitation.
Please see our answer above about this topic.

P16L496: Are these correlations significant?
The correlations are signification and the p-values have been added in the revised manuscript.

P17L538: Maybe I missed something, but there is no experiment where the sensitivity of snow drift simulations in CHM is shown without the WindNinja fields.

We use this sentence since we tested in our paper the sensitivity of the snow drift simulations to the parameterization of wind reduction in leeward areas. This sentence has been rephrased as follows:

*"Results of blowing snow redistribution simulations in CHM were sensitive to the quality of the driving wind field,* **in particular the impact of recirculation areas**, *at the mountain range scale (> 100 km$^2$)."*

References (not included in the initial manuscript):

Liston, G. E., & Elder, K. (2006). A meteorological distribution system for high-resolution terrestrial modeling (MicroMet). *Journal of Hydrometeorology*, *7*(2), 217-234.

Menke, R., Vasiljević, N., Mann, J., and Lundquist, J. K.: Characterization of flow recirculation zones at the Perdigão site using multi-lidar measurements, Atmos. Chem. Phys., 19, 2713–2723, https://doi.org/10.5194/acp-19-2713-2019, 2019.

Thornton, P. E., Running, S. W., and White, M. A. (1997). Generating surfaces of daily meteorological variables over large regions of complex terrain. *J. Hydrol.* 190, 214–251. doi: 10.1016/S0022-1694(96)03128-9

---

## Author Response (AR2)

Vincent Vionnet
Environnement and Climate Change Canada, Dorval, Canada
Tel.: +1 438 366 0148
Email: vincent.vionnet@canada.ca

December 22nd, 2020

Dear *The Cryosphere* Editor,

Please find enclosed the accepted version of the manuscript TC-2020-187. Compared to the version submitted on October 23rd 2020, we made two minor changes. As requested by R. Mott in her report, we replaced Grunewald by Grünewald at L 458 to be consistent with the name is the reference list. We also changed the reference to Pomeroy and Gray (1995) in the reference list to properly refer to the book.

Sincerely yours,

Vincent Vionnet
and co-authors.